# Self-organized canals enable long-range directed material transport in bacterial communities

Ye Li[1†], Shiqi Liu[1†], Yingdan Zhang[2], Zi Jing Seng[3], Haoran Xu[1], Liang Yang[2]*, Yilin Wu[1]*

[1]Department of Physics and Shenzhen Research Institute, The Chinese University of Hong Kong, Hong Kong, China; [2]School of Medicine, Southern University of Science and Technology, Shenzhen, China; [3]Singapore Center for Environmental Life Science Engineering, Nanyang Technological University, Singapore, Singapore

**\*For correspondence:**
yangl@sustech.edu.cn (LY);
ylwu@cuhk.edu.hk (YW)

[†]These authors contributed equally to this work

**Competing interest:** The authors declare that no competing interests exist.

**Abstract** Long-range material transport is essential to maintain the physiological functions of multicellular organisms such as animals and plants. By contrast, material transport in bacteria is often short-ranged and limited by diffusion. Here, we report a unique form of actively regulated long-range directed material transport in structured bacterial communities. Using *Pseudomonas aeruginosa* colonies as a model system, we discover that a large-scale and temporally evolving open-channel system spontaneously develops in the colony via shear-induced banding. Fluid flows in the open channels support high-speed (up to 450 μm/s) transport of cells and outer membrane vesicles over centimeters, and help to eradicate colonies of a competing species *Staphylococcus aureus*. The open channels are reminiscent of human-made canals for cargo transport, and the channel flows are driven by interfacial tension mediated by cell-secreted biosurfactants. The spatial-temporal dynamics of fluid flows in the open channels are qualitatively described by flow profile measurement and mathematical modeling. Our findings demonstrate that mechanochemical coupling between interfacial force and biosurfactant kinetics can coordinate large-scale material transport in primitive life forms, suggesting a new principle to engineer self-organized microbial communities.

## Editor's evaluation

Congratulations on a very nice study. We see the discovery of the formation of large self-organized canals within bacterial colonies as a fascinating phenomenon. Furthermore, your thorough analysis describing the capacity of bacteria to employ these canals for rapid long-distance intercellular transportation has major implications for our understanding of bacterial cooperation. We are delighted to have the opportunity to present your fundamental findings to the scientific community.

## Introduction

Long-range directed material transport is essential to maintain the physiological functions of multicellular organisms; it helps an organism to transport nutrients, metabolic wastes, and signaling molecules, translocate differentiated sub-populations through the body, and maintain pH or temperature homeostasis. Long-range directed transport in multicellular qorganisms is primarily driven by pressure-induced advection and coordinated cilia beating. For example, hydraulic pressure due to active pumping drives the circulation of body fluid in blood and lymph vessel systems of animals (*Scallan et al., 2016*); transpiration and capillary pressure passively drive the water transport through vascular tissues of plants (*Sack and Holbrook, 2006*) and cilia beating of epithelial cells drives the

cerebrospinal fluid flow in brain ventricles (*Faubel et al., 2016*) as well as mucus flow in the respiratory tract (*Huang and Choma, 2015*).

Like in animals and plants, any form of long-range directed material transport would undoubtedly bring profound effect to the development, structure, and stress response of bacterial communities. Establishing autonomous long-range material transport will be of particular importance to maintain and control the physiology of engineered functional living materials consisting of large-scale synthetic microbial consortia (*Chen et al., 2015b*; *Rodrigo-Navarro et al., 2021*). However, material transport in bacterial world is often short-ranged and limited by diffusion (either passive diffusion due to thermal energy or active diffusion due to self-propulsion of motile cells; *Wu and Libchaber, 2000*). At the single-cell level, diffusion governs nutrient uptake and sets a fundamental limit on the size of bacterial cells (*Berg, 1993*; *Nelson, 2003*; *Schulz and Jorgensen, 2001*). In bacterial communities, diffusion has been assumed to dominate material transport (*Lavrentovich et al., 2013*; *Pirt, 1967*; *Shao et al., 2017*); long-range directed material transport is deemed unusual, despite the notion that bacterial communities resemble multicellular organisms in many aspects such as coordinated metabolism, communication, and division of labor (*Lee et al., 2017*; *Parsek and Greenberg, 2005*; *Shapiro, 1998*; *van Gestel et al., 2015*).

Intriguingly, a few examples of long-range directed transport in bacterial communities were reported in recent years. Among these examples, it was shown that long-range flows were driven by flagellar motility in sediment biofilms (*Fenchel and Glud, 1998*; *Petroff and Libchaber, 2014*) and in bacterial colonies (*Wu et al., 2011*; *Xu et al., 2019*), with a typical flow speed comparable to the swimming speed of individual cells; directed transport can also be driven by passive forces such as osmosis and evaporation-induced pressure gradient (*Wilking et al., 2013*; *Wu and Berg, 2012*), with a speed of ~0.1–10 µm/s. Nonetheless, these forms of directed material transport all appear to lack autonomous regulation at the community level; the transport is either passive (*Wilking et al., 2013*; *Wu and Berg, 2012*) or driven by locally interacting cells (*Fenchel and Glud, 1998*; *Petroff and Libchaber, 2014*; *Wu et al., 2011*; *Xu et al., 2019*). Intra-colony channel structures have been identified in a few bacterial species (*Davey et al., 2003*; *Drury et al., 1993*; *Rooney et al., 2020*; *Stoodley et al., 1994*; *Xu et al., 2019*); these channel structures have a typical persistence length of a few micron to tens of microns, thus not able to support directed transport beyond millimeter scale.

In general, active or autonomous long-range directed transport would require a spatially or temporally regulated source of driving force. For bacterial communities, interfacial tension may potentially offer such a driving force because many bacterial species synthesize biosurfactants (i.e., surface active agents that reduce interfacial energy or surface tension *Chandler, 1987*) in a tightly regulated manner (*Ron and Rosenberg, 2001*). Here, we report a unique form of active long-range directed transport in structured bacterial communities enabled by spatial-temporal control of interfacial force. Using *Pseudomonas aeruginosa* colonies as a model system (*Ramos et al., 2010*), we discovered that large-scale, time-evolving open channels spontaneously emerge in the colony; these centimeter-long channels with a free surface are referred to as 'bacterial canals.' Fluid flows in the bacterial canals support high-speed (up to 450 µm/s) transport of cells and outer membrane vesicles (OMVs) over centimeters and help to eradicate colonies of a competing species *Staphylococcus aureus*. The canal flows are driven by surface tension gradient mediated by the *P. aeruginosa*-produced biofurfactant rhamnolipids, and the canals presumably emerge via a complex-fluid phenomenon known as shear-induced banding (*Divoux et al., 2016*; *Olmsted, 2008*). Overall, our findings demonstrate that mechanochemical coupling between interfacial force and biosurfactant kinetics can coordinate large-scale material transport in primitive life forms, suggesting a new principle to design macroscopic patterns and functions of synthetic microbial communities (*Brenner et al., 2008*; *Chen et al., 2015a*; *Kong et al., 2018*; *Luo et al., 2021*; *Miano et al., 2020*).

## Results

### *P. aeruginosa* colonies establish large-scale open channels supporting directed fluid transport

We cultured *P. aeruginosa* PA14 colonies on M9DCAA agar plates ('Methods'). In the colonies of wildtype and *pilB*-knockout mutant (PA14 Δ*pilB*; denoted as non-piliated mutant), we were intrigued to observe rapid cellular flows streaming through the interior of the colonies, while there was little

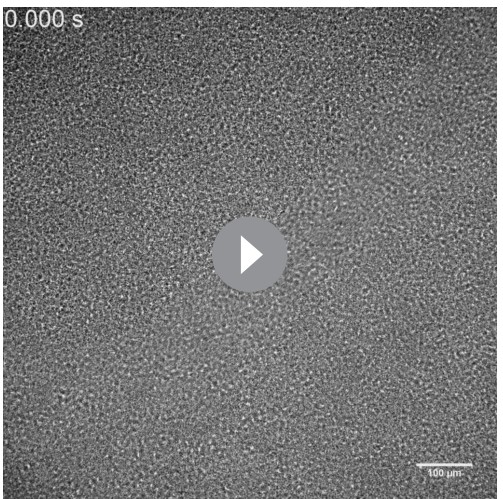

**Video 1.** Rapid cellular flow streaming through a non-piliated *P. aeruginosa* (PA14 Δ*pilB*) colony. Cells are motile outside the stream.

https://elifesciences.org/articles/79780/figures#video1

collective directed motion on both sides of the streams (*Video 1*). The streams were tens of microns in width and up to millimeters in length, but their courses were highly unstable presumably due to continuous disruption by cellular motion driven by flagellar motility. On the other hand, for PA14 mutants without flagellar motility, namely, PA14 *flgK*::Tn5 Δ*pilA* (with both flagellar motility and type IV pilus-mediated motility disabled; denoted by 'non-motile *P. aeruginosa*') (*Beaussart et al., 2014*) and PA14 *flgK*::Tn5 (without flagellar motility but with functional type IV pilus-mediated motility; denoted by 'piliated *P. aeruginosa*') (*Lee et al., 2011*; 'Methods'), we observed by naked eyes that the colonies of both strains presented many low-cell-density valleys extending from colony center to the edge over centimeters with high directional persistence (*Figure 1A–D*). Under the microscope, we found that these remarkable, centimeter-long valleys were fluid-filled, free-surface channels (i.e., open channels) ~5–10 µm in height and tens to several hundred µm in width, in which cells carried by the fluid flow streamed rapidly at speeds up to hundreds of µm/s in coherent directions (*Figure 1E*, *Appendix 1—figure 1A*, *Figure 1—video 1*). The fluid flow in channels on average went towards the colony edge and stopped abruptly at the very end (i.e., the tip) of a channel, disappearing into the dense layer of cells near the edge (*Appendix 1—figure 1B*). The fluid flow was sensitive to water content in the air environment, and it was easily disrupted by decrease of humidity. Cells translocating along the channels eventually settled in near the colony edge and they may contribute to colony expansion; however, channel formation does not necessarily coincide with colony expansion (e.g., see 'Discussion'). Nearby channels could merge with each other while individual channels could split, resulting in a large-scale channel network across the entire colony (*Figure 1B and D*). These large-scale open channels have a free upper surface, and therefore, they are distinct from the pipe-like closed channels or conduits previously reported in bacterial colonies (*Drury et al., 1993*; *Rooney et al., 2020*; *Stoodley et al., 1994*; *Xu et al., 2019*). The open channels we found here are reminiscent of human-made canals for cargo transport, so we refer to these open channels as 'bacterial canals.'

Microscopically, the bacterial canals observed in non-motile (PA14 *flgK::Tn5* Δ*pilA*) and piliated (PA14 *flgK::Tn5*) *P. aeruginosa* colonies are similar to the unstable streams observed in wildtype *P. aeruginosa* (PA14) and non-piliated mutant (PA14 Δ*pilB*) colonies, but the former have more stable courses and are thus able to support sustained long-range directed fluid transport. For this reason, and to exclude any potential contribution of flagellar motility to material transport (*Wu et al., 2011*; *Xu et al., 2019*), we focused on bacterial canals in this study. When the amount of surface water on agar plates was reduced by extended drying ('Methods'), the piliated *P. aeruginosa* (PA14 *flgK::Tn5*) colonies displayed a unique branching morphology presumably driven by fingering instability in the presence of surface tension gradient in the colony (*Trinschek et al., 2018*; *Troian et al., 1989*; *Figure 1F*). Interestingly each branch was highly directed and hosted a single canal that ran through the entire branch (*Figure 1F*, *Figure 1—video 2*; *Appendix 1—figure 1C*). The emergence of canals in such branching colonies occurred robustly at ~8 hr (at 30°C) or ~15 hr (at room temperature) after inoculation. Since canals in the branching colonies advanced in a predictable manner without merging, hereinafter we systematically characterized canal development and manipulated this process using branching colonies of piliated *P. aeruginosa* (PA14 *flgk*::Tn5).

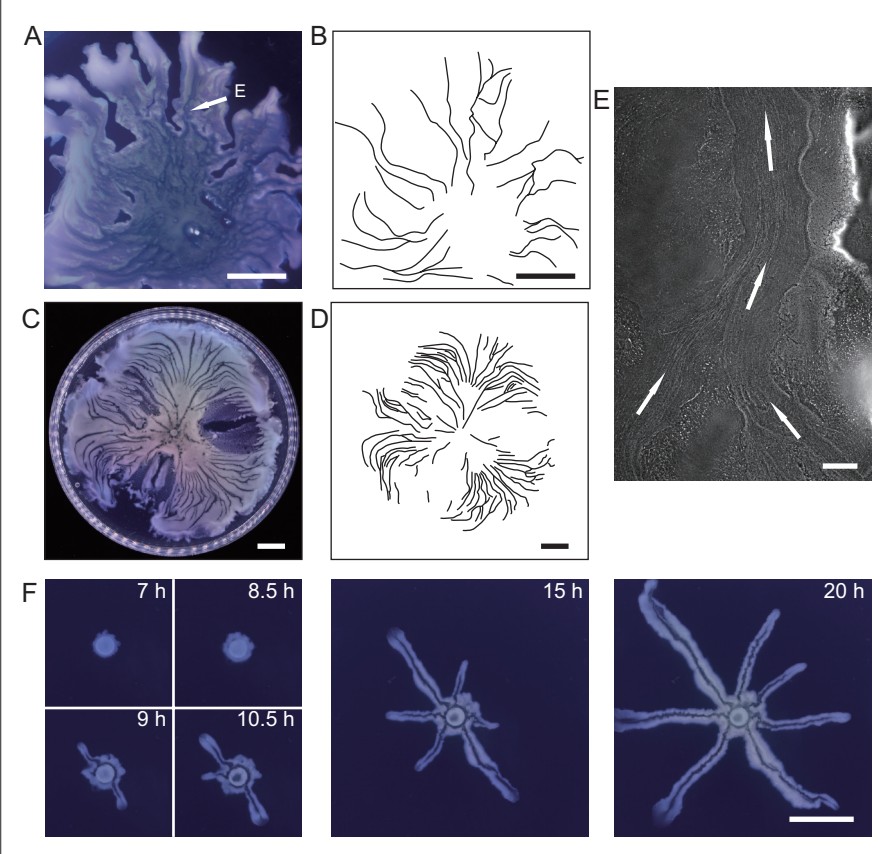

**Figure 1.** Large-scale open channels in non-flagellated *P. aeruginosa* colonies support directed fluid transport. (**A**) Colony morphology of non-motile *P. aeruginosa* (PA14 *flgK*::Tn5 Δ*pilA*). The arrow indicates the position where the image in panel (**E**) was taken. (**B**) Sketch of the open channels in panel (**A**) for better visualization. (**C**) Colony morphology of piliated *P. aeruginosa* (PA14 *flgK*::Tn5). (**D**) Sketch of the open channels in panel (**C**) for better visualization. Scale bars in A–D, 1 cm. (**E**) Phase-contrast microscopy image of an open channel with cellular flows in a non-motile *P. aeruginosa* (PA14 *flgK*::Tn5 Δ*pilA*) colony, taken at the location indicated by the arrow in panel (**A**). Arrows indicate flow direction. Scale bar, 100 μm. Cellular flows in the open channel are better visualized in *Figure 1—video 1* because the contrast between the open channel and other regions is low in still images. (**F**) Time-lapse image sequence showing the development of open channels in branching colonies of piliated *P. aeruginosa* (PA14 *flgK*::Tn5). Scale bar, 1 cm. Also see *Figure 1—video 2*.

The online version of this article includes the following video for figure 1:

**Figure 1—video 1.** Cellular flow in an open channel of non-motile *P. aeruginosa* (PA14 *flgK*::Tn5 Δ*pilA*) colony.

https://elifesciences.org/articles/79780/figures#fig1video1

**Figure 1—video 2.** Development of open channels in branching colonies of piliated *P. aeruginosa* (PA14 *flgK*::Tn5).

https://elifesciences.org/articles/79780/figures#fig1video2

## Canal development requires rhamnolipids and is driven by surface tension gradient

*P. aeruginosa* produces the well-characterized biofurfactant (i.e. surface active agents that reduce interfacial energy or surface tension; *Chandler, 1987*) rhamnolipids (*Lang and Wullbrandt, 1999*; *Müller et al., 2012*). Since canal development does not require cell motility, we hypothesized that surface tension gradient (i.e., Marangoni stress; *Davey et al., 2003*) mediated by rhamnolipids provided the driving force for fluid transport in canals; note that the canal flows cannot be driven by osmotic pressure because osmolarity gradients of cell products (hence the resultant osmotic flows) must be directing toward the colony center. As in situ measurement of rhamnolipid concentration or surface tension within colonies and canals is challenging, to examine our hypothesis, we chose to

knock out *rhlA* gene (encoding a rhamnosyltransferase essential for the production of rhamnolipids; *Ochsner et al., 1994*) in pilated *P. aeruginosa* ('Methods'). We found that this rhamnolipid-deficient mutant (PA14 *flgK*::Tn5 Δ*rhlA*) was unable to develop canals (*Figure 2A*); also the colony showed no sign of any mesoscale flows under the microscope. Next, to show that Marangoni stress could drive canal formation, we established artificial surface tension gradient in colonies by injecting an exogenous source of rhamnolipids via a programmable syringe pump ('Methods'), and it restored canal formation in colonies of the rhamnolipid-deficient mutant (*Figure 2B*). We note that the artificial surface tension gradient also promoted colony expansion (*Figure 2B*), but again canal formation does not necessarily coincide with colony expansion (see 'Discussion'). On the other hand, to examine whether canal formation involves agar degradation due to any potential agarase or hydrolase activities, we measured the height profile of the colony and agar with laser scanning confocal microscopy ('Methods'). We found that while the colony thickness above agar had an abrupt drop near canals, the height of agar under canals and other regions of the colony was uniform and there was no sign of agar degradation (*Appendix 1—figure 2*). Taken together, these results show that canal development requires rhamnolipids but not agar degradation, and they provide strong evidence that fluid flows in canals are driven by rhamnolipids-mediated surface tension gradient.

When we applied a counteracting surface tension gradient directing toward the colony center by placing an agar patch containing surfactant Tween 20 (50 mg/mL, surface tension ~4 × $10^{-2}$ N/m) ~1 cm in front of a colony branch, fluid flows in the canal gradually ceased and the canal (but not the colony branch) slowly retracted (*Video 2*; 'Methods'), suggesting that the driving force of canal flows is comparable in magnitude to that provided by a surface tension gradient of ~ $10^3$ mN·m$^{-2}$. Moreover, using a previously developed fluorescence reporter P$_{rhlA}$-*gfp*(ASV) for rhamnolipids synthesis (*Yang et al., 2009*) GFP-ASV is a short-lived derivative of GFP and its fluorescence intensity reflects the current rate of biosynthesis (*Andersen et al., 1998*; 'Methods'), we found that the *rhlA* promoter activity was azimuthally symmetric in early-stage colonies and the overall P$_{rhlA}$-*gfp*(ASV) fluorescence increased monotonically until canals emerged (*Figure 2C and D*). The result shows that rhamnolipids in the colony center were continuously synthesized, thus generating a radial gradient of surface tension.

## Flow profiles reveal shear-induced banding and surface tension distribution during canal development

The temporal dynamics of fluorescence reporter P$_{rhlA}$-*gfp*(ASV) described in *Figure 2D* indicated continuous accumulation of rhamnolipids in the colony center, which would generate a radial surface tension gradient with azimuthal symmetry at the initial stage of canal development. It is then intriguing why rapid flows only emerge in certain regions of the colony (i.e., in canals), even though at the initial stage of canal development every part of the colony at the same distance to the colony center should experience similar Marangoni stress; for instance, canals had already emerged while the colony was still nearly symmetric at T = 15 hr in *Figure 2C*. Flow speed measurement by particle image velocimetry (PIV) analysis ('Methods') in colonies before canals became visible to naked eyes revealed that the flow speed profile in regions with homogeneous cell density distribution displayed flow regimes with distinct shear rates (*Figure 3A and B*, *Figure 3—video 1*; 'Methods'). The course of those high-shear-rate domains were initially unstable (*Figure 3C*, 0–24 min) and similar to the unstable streams observed in wildtype *P. aeruginosa* (PA14) and non-piliated mutant (PA14 Δ*pilB*) colonies; as time went by, fluid flows in the high shear domains carried away cells in nearby areas, and the course got widened and gradually became fixed canals (*Figure 3C*, 32–56 min, *Figure 3—video 2*). The presence of distinct flow regimes under presumably uniform shear stress (*Figure 3A and B*) and the instability of flow courses (*Figure 3C*) are hallmarks of shear-induced banding, a phenomenon often seen in complex fluids (*Divoux et al., 2016*; *Olmsted, 2008*; *Ovarlez et al., 2009*). These results suggest that canals emerge via the onset of shear-induced banding in the colony fluids (see more in 'Discussion').

After the onset of canals, due to the coupling of rhamnolipid transport, cellular transport and quorum-sensing (QS) (*Mukherjee and Bassler, 2019*) regulation of rhamnolipid synthesis (*Lang and Wullbrandt, 1999*; *Müller et al., 2012*), rhamnolipid distribution or surface tension along the canals may vary in space and time, giving rise to a complex and dynamic profile of surface tension gradient that drives fluid flows in canals. To characterize the spatial distribution of surface tension gradient, we sought to measure the flow speed profile in canals because flow speed is linearly proportional

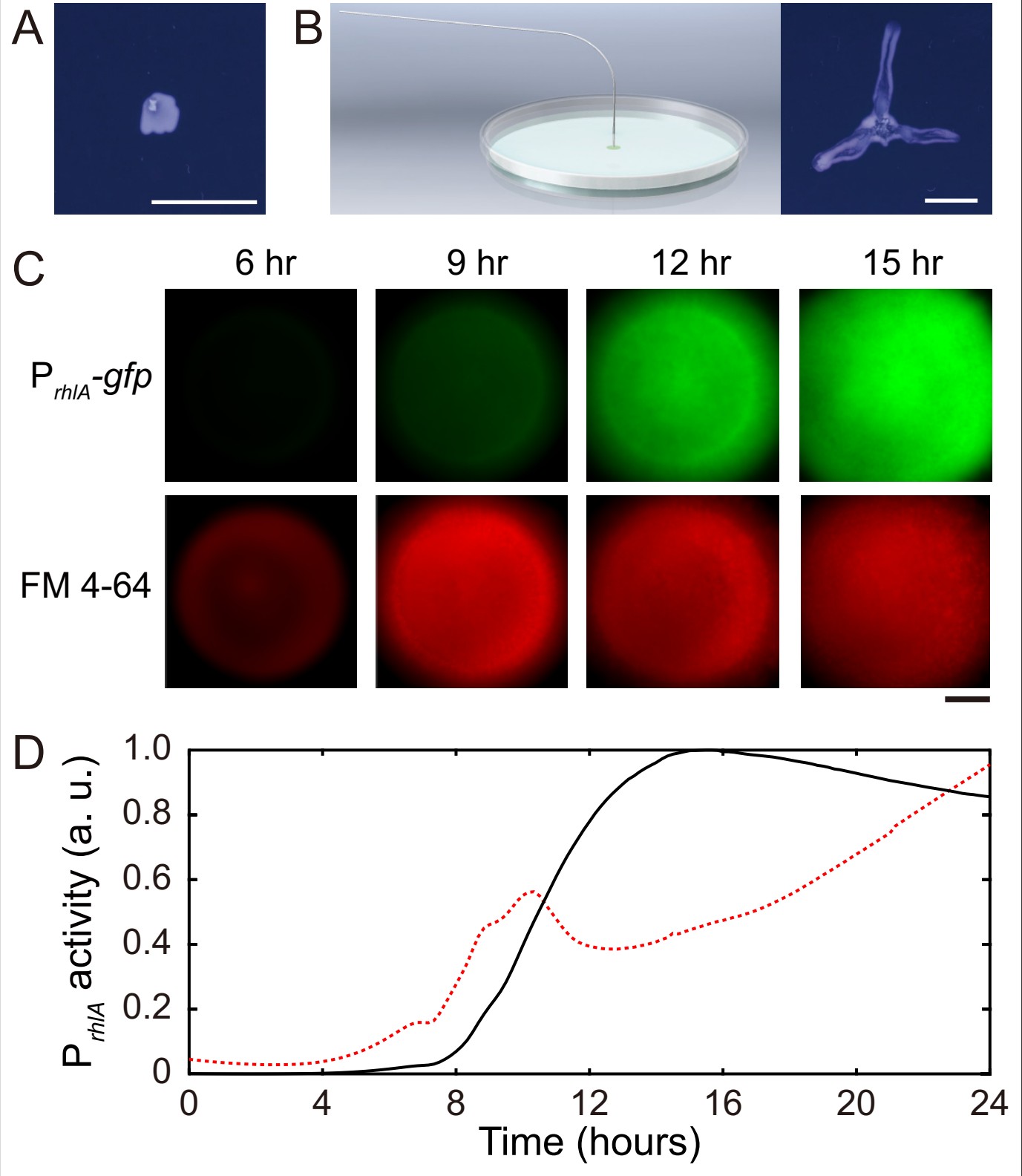

**Figure 2.** Role of rhamnolipids in canal development. (**A**) Image of a representative colony of the rhamnolipid-deficient mutant (PA14 *flgK*::Tn5 Δ*rhlA*) after 20 hr growth. The colony failed to develop canals and microscopic flows. (**B**) Surface tension gradient established by 1 hr injection of exogenous rhamnolipids at the colony center following 20 hr growth (schematics shown as left panel; 'Methods') restored the formation of canals (right panel). Scale bars in (**A**, **B**), 1 cm. (**C**) Fluorescence image sequences showing promoter activity of rhamnolipid synthesis at the early stage of canal development.

*Figure 2 continued on next page*

*Figure 2 continued*

The images were taken at the center of representative canal-forming *P. aeruginosa* colonies (PA14 *flgK*::Tn5) grown at room temperature. The upper row shows GFP(ASV) fluorescence from the rhamnolipid synthesis reporter P$_{rhlA}$-*gfp(ASV)* ('Methods'), and the lower row shows red fluorescence of the membrane dye FM 4-64, which serves as a proxy of cell number in the colony. Scale bar, 1 mm. (**D**) Temporal dynamics of rhamnolipid synthesis level measured by the fluorescence of P$_{rhlA}$-*gfp(ASV)* reporter during canal development. Solid and dashed lines represent the overall fluorescence count of the rhamnolipid synthesis reporter and the FM 4-64 dye, respectively ('Methods'). The colony started to expand at T = ~10 hr, and the expansion caused a slight drop of overall FM 4-64 fluorescence count during T = ~10–12 hr. Three replicate experiments were performed, and they showed the same temporal dynamics.

to Marangoni stress. PIV analysis as performed in *Figure 3A* can only yield the spatially averaged collective speed, so we switched to local velocity measurement with fluid tracers. In order to avoid perturbing canal flows by introducing external fluid tracers, we took advantage of the fact that some cells being transported along the canals were well isolated from others and these isolated cells could be used as natural fluid tracers. We seeded the colony with a small proportion of GFP(ASV)-expressing cells (PA14 *flgK*::Tn5 P$_{lasB}$-*gfp(ASV)*; 'Methods'), and measured the flow speed in canals by tracking the movement of these fluorescent cells. The speed of cells being transported by canals were too fast to resolve cellular positions in a single-image frame, so we computed the time-averaged speed of cells based on the long-exposure-time trajectories of cells in order to reduce the error in speed measurement (*Figure 3D*; 'Methods'). The speed of these cells varied significantly along a canal cross section; it peaked near the canal center and was attenuated near the canal boundaries. We found that the time-averaged peak cell speed near the center of canals was ~200 µm/s (with transient speeds up to 450 µm/s) (*Figure 3D*), higher than other reported forms of bacterial long-range directed transport (*Fenchel and Glud, 1998*; *Petroff and Libchaber, 2014*; *Wilking et al., 2013*; *Wu et al., 2011*; *Xu et al., 2019*) and most forms of active bacterial motility (*Mitchell and Kogure, 2006*). Note that the type IV pilus-mediated motility did not contribute to the movement of these isolated cells being transported in canals since type IV pilus-mediated motility requires surface attachment and the resultant speed is only a few µm/s (*Talà et al., 2019*), so the movement of cells indeed followed fluid motion in canals. We further measured the peak flow speed at different locations of canals. As shown by the plateau in *Figure 3E*, we found a high-flow-speed region spanning ~20 mm from the canal tip toward the colony center, and the flow speed diminished further toward the colony center. This result reveals the spatial distribution of Marangoni stress and suggests that the surface tension near the colony center has decreased to a steady-state level, presumably due to the saturated concentrations of rhamnolipids there.

## Spatial-temporal dynamics of fluid transport in canals

Measuring the speed profile as shown in *Figure 3E* along a typical ~3-cm-long canal is challenging; it requires scanning over at least ~10 locations, with each location taking ~5 min and the entire measurement taking >~1 hr ('Methods'). The trade-off between the large spatial scale of canals (centimeters) and the microscopic nature of speed-profile measurement makes it even more difficult to measure the temporal evolution of canal flows in experiment, so we resorted to mathematical modeling for understanding the spatial-temporal dynamics of canal flows. First of all, we used finite-element simulation of Navier–Stokes equation in a simplified canal geometry (*Figure 4A*; 'Methods') to estimate that the Marangoni stress in canals

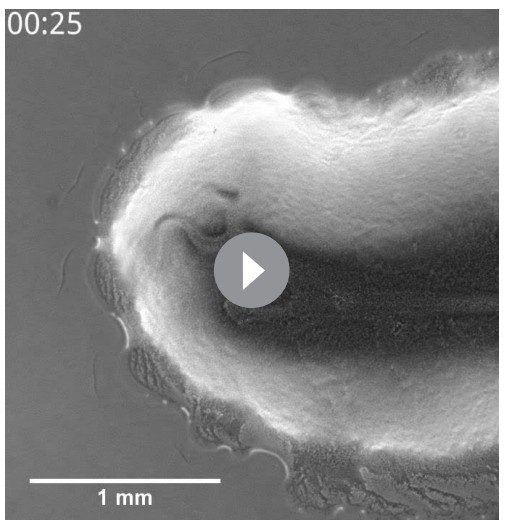

00:25

1 mm

**Video 2.** Effect of externally applied counteracting surface tension gradient on fluid transport in canals. An agar patch infused with surfactant Tween 20 (50 mg/mL) was placed at ~1 cm in front of a colony branch of piliated *P. aeruginosa* (PA14 *flgK*::Tn5); see 'Methods'. As shown in the video, immediately following this operation, fluid flows in the canal gradually ceased and the canal slowly retracted. Time label shows the elapsed time from placing the agar patch (mm:ss). https://elifesciences.org/articles/79780/figures#video2

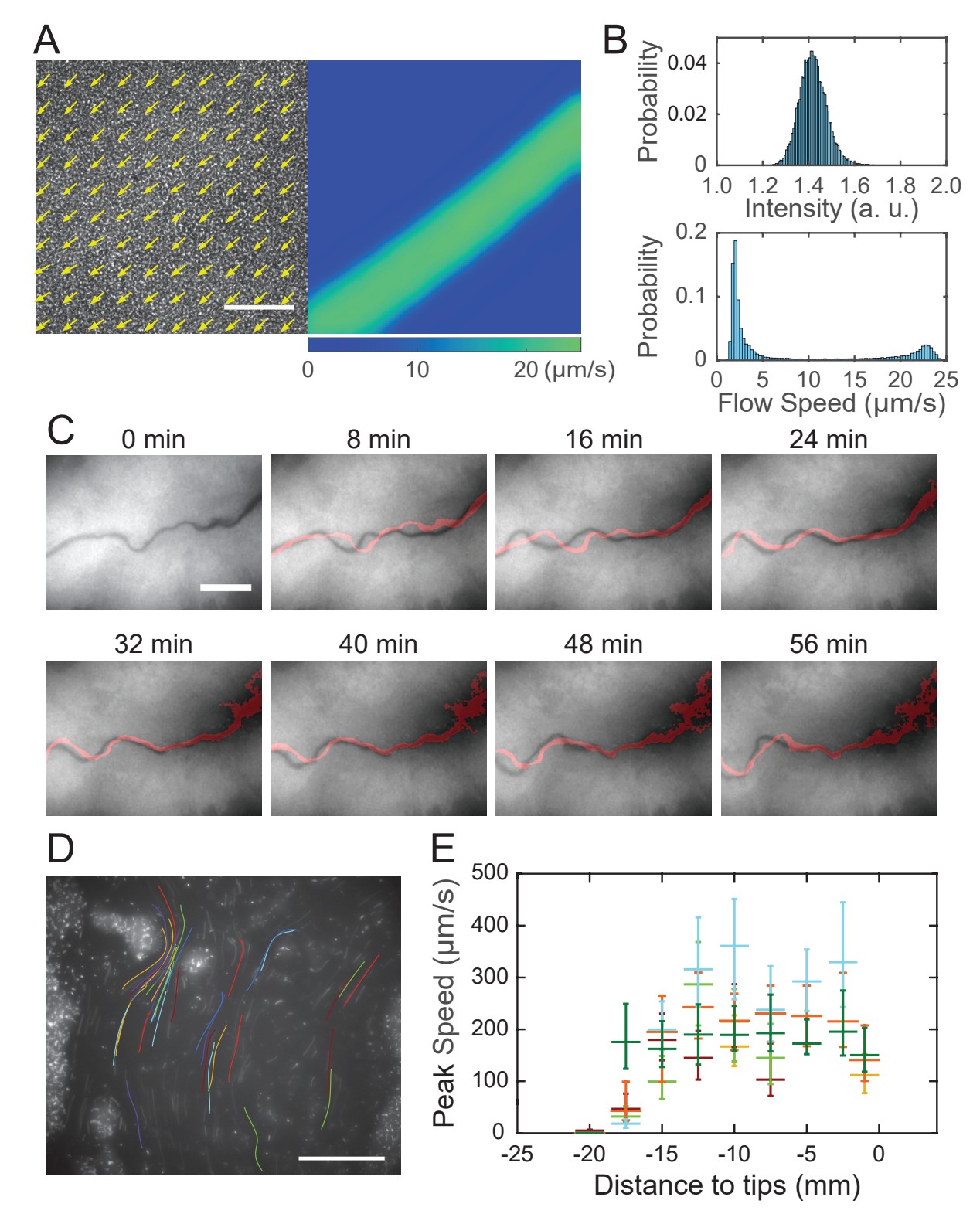

**Figure 3.** Shear-induced banding during canal development and flow speed profile in canals. (**A**) Collective velocity field of cells in a region with homogeneous cell density distribution prior to canal formation. The collective velocity field was measured by particle image velocimetry (PIV) analysis on phase-contrast image sequence and averaged over a time window of 4 s ('Methods'). Arrows in left panel and colormap in right panel represent collective velocity direction and magnitude, respectively. The velocity direction field in left panel is superimposed onto the phase-contrast image of

*Figure 3 continued on next page*

*Figure 3 continued*

this region. Both panels share the same scale bar (100 µm). Also see *Figure 3—video 1*. (**B**) Probability distributions of phase-contrast image intensity (upper panel) and flow speed (lower panel) associated with panel (**A**). Unimodal distribution of phase-contrast intensity indicates a homogeneous distribution of cell density. The bimodal distribution of flow speeds shows the occurrence of a high-speed flow regime. (**C**) Image sequence showing the course of a developing canal in a colony. Cells in the colony hosted the GFP-reporter plasmid expressing P$_{rhlA}$-*gfp(ASV)* and the colony was imaged by fluorescence microscopy ('Methods'). The course of the canal (across the center of each image) had lower cell density due to flushing by rapid flows and thus appeared darker than other areas. The red trace in each image starting from T = 8 min indicates the course of the canal in the previous image (8 min earlier). Scale bar, 125 µm. Also see *Figure 3—video 2*. (**D**) The fluorescence image of a canal in a *P. aeruginosa* colony seeded with ~1% GFP-expressing cells. The image was taken at a distance of ~12.5 mm from the tip of a canal. Colored traces show the trajectories of 38 fluorescent cells being transported in the canal, which were recorded during 0.5 s exposure time of a single image frame. Scale bar, 100 µm. Different colors serve to distinguish trajectories of different cells. (**E**) Peak flow speed at different locations of canals. The horizontal axis indicates the distance from the canal tips to the measuring position. Horizontal error bars indicate the uncertainty of canal position measurement (1 mm). Vertical error bars indicate the full range of peak speeds measured from >20 cell trajectories. Data from independent experiments are presented in different colors.

The online version of this article includes the following video for figure 3:

**Figure 3—video 1.** Distinct flow regimes occurring in a homogeneous region of a piliated *P. aeruginosa* (PA14 *flgK*::Tn5) colony.
https://elifesciences.org/articles/79780/figures#fig3video1

**Figure 3—video 2.** The course of a developing canal in a colony.
https://elifesciences.org/articles/79780/figures#fig3video2

was on the order of ~1000 mN·m$^{-2}$ in order to support a peak flow speed of ~400 µm/s. This magnitude roughly corresponds to the surface tension gradient established between a saturated source of rhamnolipids (surface tension ~30 mN/m; *Appendix 1—figure 3*) and a surfactant-free region (surface tension ~70 mN/m; *Appendix 1—figure 3*) over a distance of ~4 cm, which is consistent with the notion that the colony center has saturated concentrations of rhamnolipids as revealed in *Figure 3E*.

We then built a model to describe the mechanochemical coupling between interfacial force and biosurfactant kinetics, which involve the transport processes of colony constituents (including rhamnolipids, cell mass, QS molecules and nutrients) and QS regulation of rhamnolipid synthesis (*Figure 4B and C*; Appendix 1). Surface tension ($\gamma$) is a function of surface density ($\Gamma$) of rhamnolipids at the liquid–air interface:

$$\gamma\left(\Gamma\right) = \Pi_{max} exp\left(-A\Gamma^2/\Gamma_c^2\right) + \gamma_\infty, \tag{1}$$

where $\gamma_\infty$ is the surface tension of saturated rhamnolipid solution (at concentration >> CMC) measured by the pendant drop assay (*de Gennes et al., 2003*; *Appendix 1—figure 3*), $\Pi_{max}$ is the difference in surface tension between pure water and the saturated rhamnolipid solution (corresponding to the maximal amount of surface tension decrease due to rhamnolipids), $\Gamma_c$ is a characteristic surface density of rhamnolipids, and $A$ is a parameter relating bulk rhamnolipid concentration to its steady-state surface density (Appendix 1). The parameters $\gamma_\infty$, $\Pi_{max}$ in *Equation 1* were obtained by fitting the experimental measurement of the surface tension of rhamnolipid solutions (*Appendix 1—figure 3*; Appendix 1). We introduced dimensionless surface density ($\Gamma_n$) and bulk concentration of biosurfactant ($c_n$) as $\Gamma_n = \Gamma/\Gamma_c$ and $c_n = c/c_s$, which are coupled through the following equations:

$$\frac{\partial c_n}{\partial t} = \nabla \cdot \left(D_c \nabla c_n\right) - \left(kc_n - kA\Gamma_n^2\right) + \frac{N}{K_N+N}\alpha_R\rho\frac{B^m}{K_B^m+B^m}, \tag{2}$$

$$\frac{\partial \Gamma_n}{\partial t} = \nabla \cdot \left(D_\Gamma \nabla \Gamma_n\right) - \nabla \cdot \left[\eta_\Gamma \Gamma_n \nabla \gamma\left(\Gamma_n\right)\right] + \left(kc_n - kA\Gamma_n^2\right) \tag{3}$$

*Equation 2* describes the variation of $c_n$ due to three processes, namely, biosurfactant diffusion, biosurfactant exchange between the liquid–air interface and the bulk phase, and biosurfactant production. The biosurfactant exchange between the two phases is a key element in the model (Appendix 1); it was considered in the analysis of Marangoni flows induced by depositing surfactants into a liquid (*Hanyak et al., 2012*; *Roché et al., 2014*) but rarely in previous studies of the role of surface tension during colony development. The biosurfactant production rate is controlled by nutrient concentration ($N$), bacterial density ($\rho$), and QS signal (auto-inducer) concentration ($B$) (*Cao et al., 2016*); $N$, $B$, and $\rho$ follow another set of differential equations (Appendix 1). Similarly, *Equation 3* describes the variation of $\Gamma_n$ due to biosurfactant diffusion, advective transport of biosurfactant by fluid flows, as well as biosurfactant exchange between the liquid–air interface and the bulk phase; the advective transport

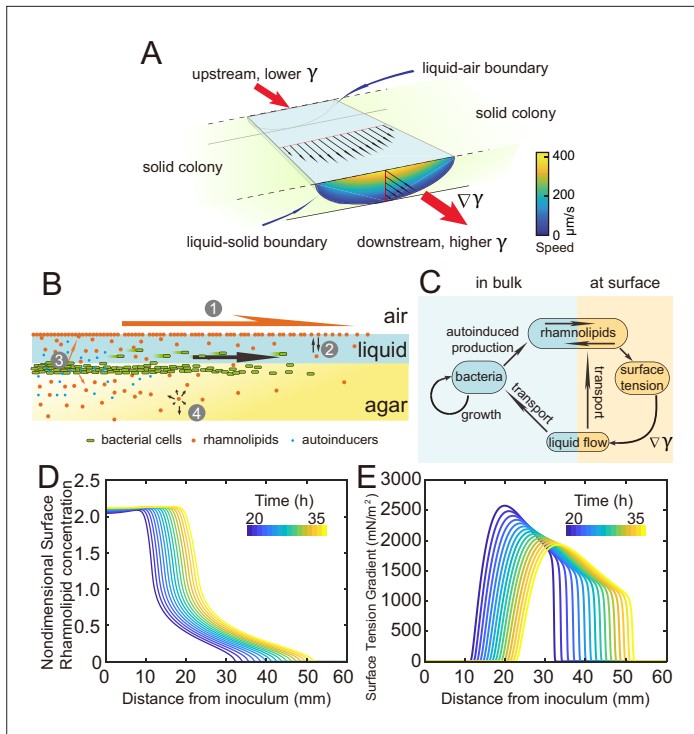

**Figure 4.** Modeling fluid transport and spatial-temporal dynamics of surface tension gradient in canals.
(**A**) Hydrodynamic simulation of fluid transport in a simplified canal geometry. The cross-sectional profile of the canal was modeled as a half ellipse with the major and minor axis being 150 μm and 10 μm, respectively. Note that the vertical and horizontal length scales are different. Red arrows indicate the direction of surface tension gradient $\nabla\gamma$. Black arrows and colormap show the surface and the bulk flow speed profiles, respectively, generated by a surface tension gradient of $1000 \text{ mN} \cdot \text{m}^{-2}$ imposed at the canal's upper surface (liquid–air boundary).
(**B**) Schematic showing the key processes involved in the establishment of surface tension gradient in canals. Major constituents of the colony are represented by symbols of different colors: surfactant molecules, orange dots; QS molecule, blue dots; cells, green rods. The key processes are labeled by numbers: '1,' transport of bacteria and biosurfactant at the liquid–air interface driven by surface tension gradient $\nabla\gamma$; '2,' biosurfactant exchange between the liquid–air interface and the bulk phase; '3,' growth of bacteria and the production of biosurfactant under QS regulation; '4,' diffusion of biosurfactant, QS molecules, and nutrient. Arrows indicate the direction of material transport. (**C**) Schematic showing the coupling of the key processes described in panel (**A**) and in main text. See Appendix 1 for details. (**D, E**) Spatio-temporal dynamics of surface-associated surfactant concentration $\Gamma$ and surface tension gradient $\nabla\gamma$ obtained by numerical simulations of the mathematical model, as shown in panels (**D**) and (**E**), respectively. Also see **Appendix 1—figure 4** for the numerical result of the spatial-temporal evolution of other quantities in the model.

term arises because biosurfactant molecules associated with the liquid–air interface are carried along by fluid flows driven by the surface tension gradient $\nabla\gamma$ (**de Gennes et al., 2003**). In **Equation 1** and **Equation 2**, $D_\Gamma$, $D_c$, $\eta_\Gamma$, $k$, $K_N$, $\alpha_R$, $K_B$, and $m$ are constant parameters. The details of model equations and parameters are described in Appendix 1 and **Appendix 1—table 2**.

As shown by the numerical simulation results based on the model, the distribution of interface-associated surfactant displayed a saturated region near the colony center (**Figure 4D**) (corresponding to the region with minimal surface tension in **Appendix 1—figure 4A**), followed by a biphasic decrease toward the spreading front of the colony (corresponding to the increase of surface tension in **Appendix 1—figure 4A**). The resultant surface tension gradient (and therefore flow speed) was nearly zero near the colony center, which agrees with the experimental result (**Figure 3E**); further away from the center, it displayed a pronounced plateau (i.e., ranging from the peak to knee of each curve at specific time points, corresponding to the region with rapid canal flows) with modest inclination (**Figure 4E**). These results are in qualitative agreement with the measured flow speed profile in canals (**Figure 3E**), justifying our model and supporting the notion that canal flows are driven by *rhamnolipids-mediated surface tension gradient*; note that the modest slope from peak to knee

shown in each curve of *Figure 4E* was not apparent in *Figure 3E*, as it could have been obscured by the large variation of flow speeds measured in experiments or by the low temporal resolution of measurement (scanning over the entire canal takes >~1 hr, during which time the true flow speed profile must have shifted and the spatial variation would be smoothed out to certain extent). More importantly, our simulation results were able to reveal that the entire plateau moves away from colony center at a speed of ~1 mm/hr, with the width increasing gradually from ~20 mm to ~30 mm in 15 hr while the height decreasing by ~20% (*Figure 4E*). Consequently, the high-speed directed flows at a specific location in the canal can persist for ~20–30 hr. Taken together, these modeling results provide a qualitative picture how a colony maintains high-speed directed transport along canals for an extended range both in space and time.

## Directed fluid flows transport outer membrane vesicles and help to eradicate competitor colonies

Finally, we examined whether the canals have other physiological functions besides helping cellular populations to translocate over long distances. *P. aeruginosa* produces OMVs to deliver pathogenic factors, antimicrobial compounds, intercellular signals, and public goods that either dissolve poorly or are prone to rapid dilution in the extracellular milieu (*Bomberger et al., 2009*; *Mashburn and Whiteley, 2005*; *Schwechheimer and Kuehn, 2015*). OMVs are slow in diffusion because their sizes are highly heterogeneous ranging from several tens to hundreds of nanometers. To examine whether directed fluid flows in canals can facilitate long-range transport of OMVs produced by *P. aeruginosa*, we isolated OMVs from bacterial culture by centrifugation, labeled them fluorescently, and loaded the OMV dispersion into a canal by microinjection ('Methods'). We found that these OMVs, having a mean size of ~150 nm (*Appendix 1—figure 5*), were transported along the canal over centimeter distances (*Figure 5A–C*) with a peak speed of >200 μm/s (*Figure 5—video 1*).

 *P. aeruginosa* is a primary member of poly-species microbial communities found in lung infections of cystic fibrosis (CF) patients, and it may engage in competitive interactions with other species such as *S. aureus* (*Chew et al., 2018*). To examine the potential function of canal flows on interspecies competition, we co-cultured colonies of piliated *P. aeruginosa* (PA14 *flgK*::Tn5) and *S. aureus* on agar plates ('Methods'). We used *S. aureus* cells harboring a plasmid with constitutive GFP expression (*Toledo-Arana et al., 2005*), so that the cell mass in *S. aureus* colonies can be quantified by GFP fluorescence during the interaction with *P. aeruginosa* ('Methods'). We found that *S. aureus* colonies irrigated by canal flows were quickly eradicated (*Figure 5D*, *Figure 5—video 2*; *Appendix 1—figure 6*), with <2% of the initial cell mass remaining after 60 min of contact with canal flows (*Figure 5E*, curve labeled as 'irrigated'). By contrast, *S. aureus* colonies that were in contact with *P. aeruginosa* colony but did not encounter canal flows retained ~40% of the initial cell mass after 60 min of contact (*Figure 5E*, curve labeled as 'non-irrigated'). These results demonstrated that fluid flows in canals help to eradicate competing bacterial species. We suggest that canal flows may increase the flux of *P. aeruginosa*-produced antimicrobial substances, such as quinolines encapsulated in OMVs with antibacterial activities against Gram-positive bacteria (*Mashburn and Whiteley, 2005*) and toxic compounds carried by rhamnolipid micelles (*Gdaniec et al., 2022*), and therefore enhance the efficiency of inhibiting competing bacterial species. It should be noted that *P. aeruginosa* produces a variety of anti-staphylococcal compounds such as quinolines, pyocyanin, and LasA protease (*Hotterbeekx et al., 2017*). While hydrophobic substances such as quinolines are poorly diffusible within the colony, those hydrophilic substances dissolvable in water readily diffuse within the agar substrate and their concentration in canals will fade out in a fraction of a second. In both cases, the anti-staphylococcal substances cannot directly benefit from canal transport, unless they are adsorbed to or encapsulated by larger particles that are slow in diffusion, such as OMVs and surfactant micelles ranging from tens to hundreds of nm in size. Therefore, the rapid eradication of *S. aureus* colonies was most likely due to substances encapsulated in OMVs (*Mashburn and Whiteley, 2005*) and rhamnolipid micelles (*Gdaniec et al., 2022*) that were transported by canals.

## Discussion

We have discovered that *P. aeruginosa* colonies develop a large-scale and temporally evolving free-surface open-channel system, which supports high-speed (up to 450 μm/s) material transport over

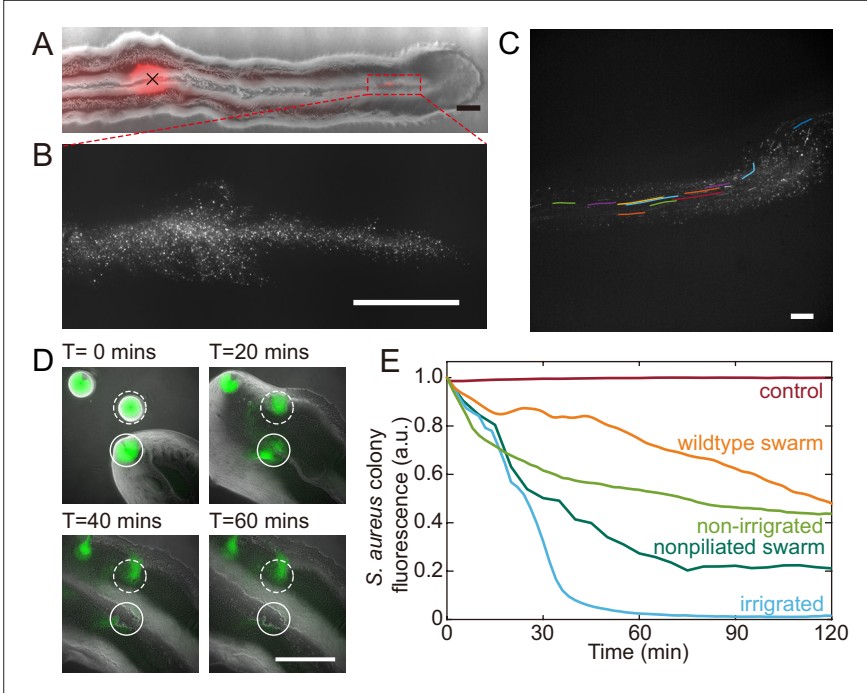

**Figure 5.** Fluid flows in *P. aeruginosa* canals transport outer membrane vesicles (OMVs) and help to eradicate *S. aureus* colonies. (**A**) Phase-contrast image of a canal overlaid with the fluorescence image of OMVs (in red). The cross sign marks the location of microinjecting OMV dispersion. The loaded OMVs were transported downstream along the canal and accumulated near the tip. Scale bar, 1 mm. (**B**) Enlarged view of the OMV fluorescence image at the red dashed box in panel (**A**). Scale bar, 500 μm. (**C**) Colored traces show the trajectories of fluorescently labeled OMVs during 1 s exposure time. Scale bar, 100 μm. Also see ***Figure 5—video 1***. (**D**) Image sequence showing the eradication of an *S. aureus* colony (GFP-labeled, enclosed by the solid line) that was irrigated by *P. aeruginosa* canal flows. A nearby *S. aureus* colony (enclosed by the dashed line) came into contact with the *P. aeruginosa* colony but did not encounter canal flows. In each panel the fluorescence image of *S. aureus* colonies is superimposed onto the phase-contrast image. Also see ***Figure 5—video 2***. (**E**) Temporal dynamics of *S. aureus* colony biomass after encountering *P. aeruginosa* PA14 colonies under different conditions. To compare with the effect of potential material transport by flagellar motility (***Wu et al., 2011***; ***Xu et al., 2019***), curves labeled as 'wildtype swarm' and 'non-piliated swarm' were shown for *S. aureus* colonies encountering wildtype *P. aeruginosa* (with both flagellar and type IV pilus motilities) and non-piliated *P. aeruginosa* (PA14 Δ*pilB*; with flagellar motility alone), respectively (see ***Appendix 1—figure 6*** for details; 'Methods'). At least three replicates were performed for each condition.

The online version of this article includes the following video for figure 5:

**Figure 5—video 1.** Transport of outer membrane vesicles (OMVs) along a canal.
https://elifesciences.org/articles/79780/figures#fig5video1

**Figure 5—video 2.** Eradication of an *S. aureus* colony irrigated by *P. aeruginosa* canal flows.
https://elifesciences.org/articles/79780/figures#fig5video2

---

centimeters. The open channels, or 'bacterial canals,' are presumably driven by rhamnolipid-mediated surface tension gradient. The spatial-temporal dynamics of fluid flows in bacterial canals are revealed by flow profile measurement and mathematical modeling that involves mechanochemical coupling between interfacial force and biosurfactant kinetics. Our findings present a conceptual advance regarding the potential role of biosurfactant-empowered Marangoni stress: it can act as a spatio-temporally regulated driving force for active long-range directed material transport within structured bacterial communities.

Many bacterial species isolated to date synthesize biosurfactants (***Ron and Rosenberg, 2001***). In surface-associated bacterial communities where interfaces are prevalent (***OToole and Wong, 2016***; ***Persat et al., 2015***), the inhomogeneous synthesis of biosurfactants often result in surface tension gradients (Marangoni stress). With the advance of synthetic biology, mechanochemical coupling

between interfacial force and biosurfactant kinetics (including the synthesis, diffusion, and transport) could become a new strategy to design self-organization and sensory functions in synthetic microbial communities (*Brenner et al., 2008*; *Chen et al., 2015a*; *Kong et al., 2018*; *Miano et al., 2020*). For instance, canal transport driven by Marangoni stress can be engineered to help translocate cellular populations and slowly diffusing chemical substances over long distances, shaping the population structure and mediating stress response of synthetic bacterial communities; the mechanism could also be exploited to transport drugs and chemical effectors loaded in vesicles in order to control the communication and behavior of synthetic bacterial communities.

Can canal transport be found in natural or clinical settings? Imaging of *P. aeruginosa* aggregation patterns in situ in sputum samples from CF patients did not reveal canal-like structures (*DePas et al., 2016*), suggesting that canal formation may not occur in such conditions. Indeed, wildtype *P. aeruginosa* colonies displayed rapid streams within the colony but the streams did not turn into stable and persistent canals (*Video 1*). The instability was presumably caused by continuous displacement of cells driven by flagellar motility. Flagellar motility weakens cell–cell adhesion and disrupts the course of high-shear flow regimes, thus preventing the high-shear flow regimes from further developing into more stable courses or canals. Although *P. aeruginosa* cells tend to downregulate flagellar motility in the airways of patients with CF (*Luzar et al., 1985*; *Mahenthiralingam et al., 1994*; *Wolfgang et al., 2004*), the residual flagellar motility could still impede the full development of canals. We note that another flagellated and biosurfactant-producing bacteria *Serratia marcescens* is able to develop canal-like structures during swarming (*Appendix 1—figure 7*), but such structures are only found at the center of the colony where most cells appear to be sessile. Taken together, observing canal formation in natural and clinical settings may require species-dependent or very specific conditions.

Marangoni stress was previously suggested to contribute to the expansion of bacterial colonies (*Angelini et al., 2009*; *Du et al., 2012*; *Fauvart et al., 2012*; *Rhodeland et al., 2020*; *Trinschek et al., 2018*). In this connection, we emphasize the difference between directed material transport within bacterial colonies and colony expansion. Colony expansion normally does not lead to directed transport of materials within the colony. For example, during swarming of wildtype *P. aeruginosa* facilitated by Marangoni stress, the colony expansion speed was ~1 μm/s (*Fauvart et al., 2012*), which is one order of magnitude slower than individual cells' random motion (tens of μm/s), and consequently, a cell or a passive tracer placed in the colony is expected to undergo diffusive motion with a small drift (*Zuo and Wu, 2020*) rather than directed transport. Moreover, as demonstrated in this study, directed material transport within a colony does not necessarily coincide with colony expansion (e.g., see *Appendix 1—figure 1—video 2*), and the speed of the two processes could differ by orders of magnitude (~200 μm/s versus ~2 mm/hr). Nonetheless, our work provides a new ingredient (i.e., long-range material transport driven by interfacial mechanics) that will complement existing models such as fingering instability (*Trinschek et al., 2018*) and colony-growth optimization (*Luo et al., 2021*) to explain and control pattern formation in expanding *P. aeruginosa* colonies.

Finally, the presence of distinct flow regimes and the instability of flow courses at the early stage of canal development (*Figure 3A and B*) suggest that canals emerge via the onset of shear-induced banding in the colony fluids. This notion points to a novel biological function of shear-induced banding that often occurs in complex fluids such as polymer solutions, surfactant micellar solutions, and colloidal suspensions (*Divoux et al., 2016*; *Olmsted, 2008*; *Ovarlez et al., 2009*). To further examine the conditions of shear banding in colony fluids, we measured the flow profiles of bacteria–rhamnolipids mixtures resembling the colony constituents in PDMS microfluidic channels subject to uniform shear stress. Under controlled flow conditions in PDMS microfluidic channels, we found that the cross-sectional flow velocity profile of a mixture of dense bacterial suspension (~$1.35 \times 10^{10}$ cells/mL) and concentrated rhamnolipids at 10 mg/mL, ~100 times critical miceller concentration (CMC) (*Zhao et al., 2018*) was highly asymmetric and signaled shear banding (*Appendix 1—figure 8A*), whereas the flow velocity profile of pure cell suspension at the same density or of a mixture of rhamnolipids (10 mg/mL) and dense 1.1 μm microsphere suspension (~$1.9 \times 10^{10}$ particles/mL) was closer to the parabolic profile characteristic of Newtonian fluids (*Appendix 1—figure 8B and C*). This result suggests that the onset of shear banding in *P. aeruginosa* colonies preceding canal formation requires the interaction between cells and rhamnolipids (most likely in the form of micelles). The detailed biophysical mechanism of this requirement warrants further investigation as a unique rheological behavior of bacterial active matter (*Guo et al., 2018*; *Liu et al., 2021*; *Martinez et al., 2020*).

# Materials and methods

**Key resources table**

| Reagent type (species) or resource | Designation | Source or reference | Identifiers | Additional information |
|---|---|---|---|---|
| Strain, strain background (*Pseudomonas aeruginosa*) | Wildtype PA14 | Roberto Kolter, Harvard University (*Lee et al., 2011*) | ZK3436 | Having both flagellar and type IV pilus motilities |
| Strain, strain background (*P. aeruginosa*) | Piliated *P. aeruginosa* PA14 | Roberto Kolter, Harvard University (*Lee et al., 2011*) | ZK3367 | Without flagellar motility but retaining type IV pilus motility |
| Strain, strain background (*P. aeruginosa*) | Non-piliated *P. aeruginosa* PA14 | Roberto Kolter, Harvard University (*Lee et al., 2011*) | ZK3351 | Without type IV pilus motility but retaining flagellar motility |
| Strain, strain background (*P. aeruginosa*) | Non-motile *P. aeruginosa* PA14 | George A. O'Toole, Dartmouth College (*Beaussart et al., 2014*) | SMC6589 | Without either flagellar or type IV pilus motility |
| Strain, strain background (*P. aeruginosa*) | *P. aeruginosa* deficient in rhamnolipid production Δ*rhlA* | This paper | PA14 *flgK*::Tn5 Δ*rhlA* | Deficient in rhamnolipid production |
| Strain, strain background (*Staphylococcus aureus*) | *Staphylococcus aureus* clinical strain that expresses GFP constitutively | *Toledo-Arana et al., 2005* | | Isolate 15981 harboring a plasmid pSB2019-*gfp* expresses GFP constitutively |
| Strain, strain background (*Serratia marcescens*) | *Serratia marcescens* | ATCC | ATCC 274 | |
| Sequence-based reagent | 1-rhlA_UpF | This paper | PCR primers | agctcggtacccggg GGGTGATTTCCTACGGGGTG |
| Sequence-based reagent | 2-rhlA_UpR | This paper | PCR primers | CTTCGCAGGTCAAGGGTTC ACCGCATTTCACACCTCCCAA |
| Sequence-based reagent | 3-rhlA_DownF | This paper | PCR primers | TTGGGAGGTGTGAAATGCGG TGAACCCTTGACCTGCGAAG |
| Sequence-based reagent | 4-rhlA_DownR | This paper | PCR primers | cgacggccagtgcca CCGTACTTCTCGTGAGCGAT |
| Sequence-based reagent | rhlA_F | This paper | PCR primers | GACAAGTGGATTCGCCGCA |
| Sequence-based reagent | rhlA_R | This paper | PCR primers | TTGAACTTGGGGTGTACCGG |
| Sequence-based reagent | rhlAGENE_F | This paper | PCR primers | GGTCAATCACCTGGTCTCCG |
| Sequence-based reagent | rhlAGENE_R | This paper | PCR primers | GCTGATGGTTGCTGGCTTTC |
| Sequence-based reagent | Pk18-F | This paper | PCR primers | TGCTTCCGGCTCGTATGTTG |
| Sequence-based reagent | Pk18-R | This paper | PCR primers | GCGAAAGGGGGATGTGCTG |
| Recombinant DNA reagent | Plasmid pMHRA | *Yang et al., 2009* | Plasmid | Contains an RlhR-regulated P$_{rhlA}$-*gfp(ASV)* fusion inserted into the vector pMH391 |
| Chemical compound, drug | Q5 High-Fidelity DNA Polymerase | NEB, USA | Cat# M0515 | |
| Chemical compound, drug | Gibson Assembly Master Mix | NEB, USA | Cat# E2611 | |
| Chemical compound, drug | Taq Polymerase | Thermo Fisher, USA | | |

## Bacterial strains and plasmids

### Bacterial strains

The following strains were used: wildtype *P. aeruginosa* PA14 (having both flagellar and type IV pilus motilities), piliated *P. aeruginosa* (PA14 *flgK*::Tn5; without flagellar motility but retaining type IV pilus motility), and non-piliated *P. aeruginosa* (PA14 Δ*pilB*; without type IV pilus motility but retaining flagellar motility), gifts from Roberto Kolter, Harvard University (*Lee et al., 2011*); non-motile *P. aeruginosa* (PA14 *flgK*::Tn5 Δ*pilA*; without either flagellar or type IV pilus motility) (*Beaussart et al., 2014*),

a gift from George A. O'Toole, Dartmouth College; *P. aeruginosa* deficient in rhamnolipid production (PA14 *flgK*::Tn5 Δ*rhlA*; see details of strain construction below); *S. aureus* clinical strain isolate 15981 harboring a plasmid pSB2019-*gfp* that expresses GFP constitutively (**Toledo-Arana et al., 2005**); *S. marcescens* ATCC 274. Single-colony isolates were grown overnight (~13–14 hr) in 10 mL culture tubes (unless otherwise stated) with shaking in LB medium (1% Bacto tryptone, 0.5% yeast extract, 0.5% NaCl) at 30°C to stationary phase. Overnight cultures were used for inoculating colonies on agar plates.

## Strain construction

Primers used are listed in *Appendix 1—table 1*. To construct the rhamnolipid-deficient strain *P. aeruginosa* PA14 *flgK*::Tn5 Δ*rhlA,* upstream of the *rhlA* gene were amplified with 1-*rhlA*_UpF and 2-*rhlA*_UpR primers. Downstream of *rhlA* gene are amplified with 3-*rhlA*_DownF and 4-*rhlA*_DownR primers. The sequences of *rhlA* were obtained from *Pseudomonas* genome database (http://www.pseudo-monas.com/). The PCR amplification was performed with Q5 High-Fidelity DNA Polymerase (NEB, USA). Using Gibson Assembly Master Mix (NEB), the two purified flanking PCR fragments (Promega, USA) were assembled with BamHI and HindIII-digested PK18 (Gm$^r$) suicide vector. 10 μL of the assembled products was transformed to *Escherichia coli* DH5a competent cell by heat shock. Transformants were selected on 60 μg/mL Gm-infused LB Lennox agar plates and the insert size was verified by PCR with Pk18-F and Pk18-R primers using Taq Polymerase (Thermo Fisher, USA). A triparental mating was performed with PA14 *flgK::*Tn5 to generate PA14 *flgK*::Tn5 Δ*rhlA* gene deletion, through conjugation together with the aid of RK600 helper plasmid strain. ABTC agar containing 10% sucrose was used for SacB-based counterselection. Mutants were confirmed by *rhlA*_F and *rhlA*_R primers with PAO1 gDNA as a control. *rhlA*GENE_F and *rhlA*GENE_R primers were used to check for the presence of *rhlA* gene with PAO1 gDNA as a control.

## Plasmids

The rhamnolipid synthesis reporter plasmid pMHRA contains an RlhR-regulated P$_{rhlA}$-*gfp*(ASV) fusion inserted into the vector pMH391 (**Yang et al., 2009**). *rhlA* encodes a rhamnosyltransferase essential for the production of rhamnolipid under the control of *rhl* QS system (**Ochsner et al., 1994**), and is thus a good indicator of *rhl* activity (**Yang et al., 2009**). GFP-ASV is a short-lived derivative of GFP, which degrades with a half-life <1 hr and is rapidly cleared from cells, thus its fluorescence intensity reflects the current rate of biosynthesis (**Andersen et al., 1998**). For flow speed measurement in canals, the plasmid pMHLB containing a translational fusion of the P$_{lasB}$-*gfp*(*ASV*) was used (**Hentzer et al., 2002**); *lasB* encodes a virulence factor elastase under the control of LasR (**Hentzer et al., 2002**) and is homogeneously expressed in the colony with sufficiently bright fluorescence suitable for our tracking method. The plasmids were introduced into PA14 *flgK*::Tn5 by electroporation. The transformants were selected on LB 1.5% agar plates supplemented with gentamicin (50 μg/mL). These plasmids have high copy numbers, so we did not need to add any antibiotics into culture environment to prevent plasmid loss.

## Preparation of agar plates for colony growth

*P. aeruginosa* colonies were grown on 0.5% Difco Bacto agar plates infused with M9DCAA medium (**Tremblay and Déziel, 2010**) (20 mM NH$_4$Cl, 12 mM Na$_2$HPO$_4$, 22 mM KH$_2$PO$_4$, 8.6 mM NaCl, 1 mM MgSO$_4$, 1 mM CaCl$_2$, 11 mM dextrose, and 0.5% [wt/vol] casamino acids [BD Bacto, Cat# 223050]). We varied the agar concentration and found that canals develop robustly over a relatively wide range of agar concentration between 0.5% and 1.0% wt/vol (*Appendix 1—figure 9*). Above 1.0%, canals were not observed presumably because the cells have difficulty to extract sufficient water to maintain an open-surface liquid film on the agar surface, which is the same underlying reason why most bacteria cannot swarm on agar above 1.0% as swarming also requires maintaining open-surface liquid films (**Kearns, 2010**). As Ca$^{2+}$ cannot coexist stably with many ions, this medium was prepared and stored in two components: (1) 10× nutrient solution without CaCl$_2$, sterilized and stored at room temperature; (2) agar infused with CaCl$_2$ at $1\frac{1}{9}$ times of the desired concentrations, sterilized and stored in 100 mL aliquots. Before use, the component (2) was melted completely in a microwave oven and cooled to ~60°C. For each plate, 18 mL molten component (2) was mixed with 2 mL component (1) and the mixture was poured to a polystyrene petri dish (90 mm diameter, 15 mm height). To quantify cell mass

in the colony, the membrane dye FM 4–64 (Thermo Fisher Scientific) was added into the agar plates at a final concentration of 1 µg/mL. The plate was swirled gently to ensure surface flatness, and then cooled for 10 min without a lid inside a large Plexiglas (PMMA) box, followed by further drying under laminar airflow for 5 min or 20 min (to culture branching colonies). 1 µL overnight culture was used to inoculate colonies at the center of each plate, and the plates were incubated at specified temperatures and >~95% relative humidity; lowering the humidity will lower the probability of canal formation. The colonies expanded at a steady-state speed of ~2 mm/hr at 30°C. The emergence of canals in branching colonies of piliated *P. aeruginosa* (PA14 *flgK*::Tn5) occurred robustly at ~8 hr (at 30°C) or ~15 hr (at room temperature) after inoculation. In experiments co-culturing *P. aeruginosa* and *S. aureus*, half of casamino acids in the M9DCAA medium described above was replaced by peptone at a final concentration of 0.25% (wt/vol) in order to support growth of *S. aureus*; this medium was referred to as M9DCAAP. The M9DCAAP plates were prepared following the same procedures as described above and incubated at 30°C and >~95% relative humidity.

## Imaging colonies and canals

The macroscale dynamics of *P. aeruginosa* colony or canal development was monitored and measured in a custom-built imaging incubator made of PMMA (l × d × h, 1 × 1 × 1.2 m). The agar plates were sealed with parafilm before incubation in order to maintain saturated humidity. The temperature of the incubator was maintained at 30°C with a heater controlled by feedback circuits. The inner walls of the incubator were covered with black cloth and the plate was illuminated by an LED strip lining at the bottom part of the side walls. The images of colonies or canals were photographed by a digital single-lens reflex camera (DSLR; Canon 700d) every 5 min during incubation (24 mm for whole plate view or 60 mm for zoomed-in view, aperture f/8, exposure time 1/5 s). The time-lapse imaging and LED illumination were triggered by a custom-programmed microcontroller (Arduino). Phase-contrast microscopy imaging of canal development was performed on an inverted microscope (Nikon TI-E) via a ×10 dark phase objective (N.A. 0.25) or a ×4 dark phase objective (N.A. 0.13). Recordings were made with an sCMOS camera (Andor Zyla 4.2 PLUS USB 3.0; Andor Technology) or the DSLR camera (Canon 700d; triggered by a custom-programmed Arduino microcontroller).

## Height profile measurement of the colony and agar with laser scanning confocal microscopy

Molten M9DCAA 0.6% agar was mixed with 0.5-µm-diameter fluorescent microsphere (FluoSpheres carboxylate-modified, ex/em: 580 nm /605 nm; Thermo Fisher F8812) at a final microsphere number density of $3.1 \times 10^{11}$ particles/mL (~1000-fold dilution of the microsphere solution). 20 mL of the agar-microsphere mixture was poured into a 60 mm diameter Petri dish. After the agar solidified, the plate was dried under laminar flow without the lid for 30 min. 1 µL overnight culture of *P. aeruginosa* PA14 *flgK*::Tn5 harboring the pMHLB plasmid (with $P_{lasB}$-*gfp(ASV)*, as described above) was inoculated into the center of the agar plate. The plate was then sealed by parafilm and incubated in a 30°C incubator without humidity control for ~24 hr before observation. The 3D structures of the agar and the bacterial colony near canals were imaged with a Leica SP8 laser scanning confocal microscope via a ×10 objective (N.A. 0.30). Green fluorescence acquired from cells in the colony (excitation: 476 nm; emission: 480–550 nm) and red fluorescence from the microspheres homogeneously dispersed in agar (excitation: 562 nm; emission: 588–658 nm) provided spatial information of the colony and the agar, respectively. The thickness of optical section was set as 12.85 µm. A total of 71 frames at different vertical positions with an interval of 4.28 µm were scanned to acquire the 3D structures. To compute the height profiles of the colony and the agar, we first obtained the 2D cross-sectional fluorescence intensity distributions in both green and red channels by averaging the 3D fluorescence image over the thickness of the optical section. We then binarized the 2D fluorescence intensity distributions in both channels to determine the boundaries of the colony and the agar, thus obtaining the thickness of the colony and the height profile the agar underneath the colony.

## Establishing exogenous surface tension gradients

Rhamnolipid solution (10 mg/mL) was injected into the colony center of the rhamnolipid-deficient mutant *P. aeruginosa* PA14 *flgK*::Tn5 Δ*rhlA*, in order to create a surface tension gradient pointing radially outward. To inject exogenous rhamnolipid solutions into the colony, 1 µL of PA14 *flgK*::Tn5 Δ*rhlA*

overnight culture was inoculated at the center of an M9DCAA agar plate, and the plate was incubated for 20 hr at room temperature and >~95% relative humidity. Then the lid was replaced by one with a drilled hole, and 10 mg/mL rhamnolipid solution (rhamnolipids 90%, in solid form, AGAE Technologies; dissolved in M9DCAA medium) loaded in a 100 μL glass microsyringe (W-018107, Shanghai Gauge) was pumped into the center of the colony at the rate of 5 μL/hr by a syringe pump (Fussion 200, Chemyx) via a PTFE tube (inner diameter 0.3 mm) that passed through the hole in the lid. The chosen rhamnolipid concentration 10 mg/mL, ~100 times CMC (*Zhao et al., 2018*) was on the order of maximal rhamnolipid concentration that can be produced by *P. aeruginosa* (*Soares Dos Santos et al., 2016*). The end of the PTFE tube barely touched the upper surface of the colony surface. Images of the colony were taken with a DSLR camera (Canon 700d).

To apply a counteracting surface tension gradient directing toward the colony center, we placed an agar patch containing surfactant Tween 20 (50 mg/mL) in front of a colony branch. To prepare the surfactant-infused agar patches, we supplemented the surfactant Tween 20 (Sigma) to molten M9DCAA 3% agar at a final concentration 50 mg/mL, poured 10 mL of this agar into 90 mm Petri dishes, and cut out 10 mm × 5 mm rectangular patches after the agar solidified. We placed a piece of the surfactant-infused agar patch at ~1 cm in front of a branch of a *P. aeruginosa* PA14 *flgK*::Tn5 colony that had grown for 20 hr at 30°C. Immediately following the placement of the agar patch, the agar plate was transferred to the stage of an inverted microscope (Nikon Ti-E) and the colony branch was observed via a ×4 dark phase objective (N.A. 0.13). Image recordings were made with an sCMOS camera (Andor Zyla 4.2 PLUS USB 3.0; Andor Technology). Control experiments followed the same procedures, except that Tween 20 solution was replaced by M9DCAA medium when preparing the agar patch.

## Fluorescence imaging of rhamnolipid synthesis reporter and data processing

We measured the synthesis level of rhamnolipids during the early stage of canal development using a fluorescence reporter $P_{rhlA}$-*gfp*(ASV) for the promoter activity of rhlA gene as described above. 1 μL of the overnight culture of *P. aeruginosa* PA14 *flgK*::Tn5 hosting the plasmid $P_{rhlA}$-*gfp*(ASV) was inoculated at the center of an M9DCAA agar plate, and the plate was incubated in a sealed glass box with four beakers (each containing 5 mL DI water) at the corners to maintain saturated humidity. The glass box was then transferred to the motorized stage (HLD117, PRIOR Scientific Instruments) on an inverted microscope (Nikon Ti-E) for imaging at room temperature. Multichannel fluorescent images were acquired with a ×4 objective (N.A. 0.13) and an FITC/Texas Red dual-band filter set (excitation: 468/34 nm and 553/24 nm; emission: 512/23 nm and 630/91 nm; dichroic: 493 to ~530 nm and 574 to ~700 nm; Semrock, Cat# GFP/DsRed-A-000) with the excitation light provided by a high-power LED light source (X-Cite XLED1, Excelitas Technologies Corp.). For excitation of GFP, BDX LED lamp module (450–495 nm) was turned on and the exposure time was set as 500 ms. For FM 4-64 excitation, GYX module (540–600 nm) was turned on and the exposure time was set as 5 s. The imaging protocol was executed by a custom-programmed microcontroller (Arduino) through the software NIS Element AR (v. 4.51, Nikon). Images were recorded with an sCMOS camera (Andor Zyla 4.2 PLUS USB 3.0; Andor Technology).

The overall fluorescence count of $P_{rhlA}$-*gfp*(ASV) reporter and FM 4-64 at the colony center was computed by integrating the fluorescence intensity over the entire inoculum region. For each image stack, the image with the lowest overall fluorescence count was chosen as the background and subtracted from the rest of images in the stack. Different colonies may have different lag times due to slight variation of inoculum size across experiments, but their growth dynamics after the lag time is highly conserved.

## Measurement of shear banding

To observe shear banding in colonies, *P. aeruginosa* PA14 *flgK*::Tn5 hosting the plasmid $P_{rhlA}$-*gfp*(ASV) was inoculated as described above. To observe shear banding in PDMS microfluidic channels, *P. aeruginosa* PA14 *flgK*::Tn5 was grown in 250 mL glass flasks with shaking in LB medium (1% Bacto tryptone, 0.5% yeast extract, 0.5% NaCl) at 30°C to a cell density ~2.1 × 10⁹ cells/mL, and the culture was concentrated to ~2.1 × 10¹⁰ cells/mL and resuspended in M9DCAA medium before use. PDMS chips with microfluidic channel was fabricated by standard soft lithography technique

(*Whitesides et al., 2001*) and sealed to cleaned glass coverslip. The cross section of microfluidic channels was rectangular, 1.1 mm in width, 200 μm in height, and 4 cm in length. Fluids (~200 μL) were loaded into a glass syringe and pumped into the microfluidic channels by the syringe pump (Chemyx Fusion 200, Two-channel Syringe Pump Model 07200) via a polytetrafluoroethylene (PTFE) tube (inner diameter 0.41 mm, outer diameter 0.92 mm). Before loading the fluids of interest, fresh M9DCAA medium was preloaded into the channel. For experiments with cell-rhamnolipid mixtures, the concentrated bacterial culture (~2.1 × $10^{10}$ cells/mL resuspended in M9DCAA medium) was mixed with rhamnolipid stock solution (100 mg/mL) at 9:1 ratio (final concentration of rhamnolipids: 10 mg/mL, ~100 times CMC; final cell density: ~1.9 × $10^{10}$ cells/mL) and injected into the microfluidic channel for observation. For experiments with microsphere-rhamnolipid mixtures, microsphere dispersion (Polybead sulfate, 1.1 μm diameter, Polyscience Inc) was mixed with rhamnolipid (final concentration of rhamnolipids: 10 mg/mL; final number density of microspheres: ~1.9 × $10^{10}$ particles/mL). The flow rate was maintained constant during observation, ranging from 3 μL/min to 5 μL/min.

The observations were performed on an inverted phase-contrast microscope (Nikon TI-E). Shear banding in colonies was observed with a ×20 objective (N.A. 0.45, w.d. 8.2–6.9 mm; Nikon, MRH48230) and experiments in PDMS microfluidic channels was imaged with a ×10 objective (N. A. 0.25, w.d. 6.2 mm; Nikon). Videos were recorded with an sCMOS camera (Andor Zyla 4.2 PLUS USB 3.0; Andor Technology) at 50 fps. Velocity field was computed by performing PIV analysis on phase-contrast microscopy image sequences using an open-source package MatPIV 1.6.1 written by J. Kristian Sveen (https://www.mn.uio.no/math/english/people/aca/jks/matpiv/). For each pair of consecutive images, the interrogation-window size started at 41.6 μm × 41.6 μm and ended at 10.4 μm × 10.4 μm after four iterations. The grid size of the resulting velocity field was 5.2 μm × 5.2 μm.

## Measurement of flow speed in canals

In order to avoid perturbing canal flows by introducing external fluid tracers, we used fluorescent cells being transported by canal flows as natural fluid tracers. Note that the type IV pilus motility did not contribute to the movement of these cells since type IV pilus motility requires surface attachment and the resultant speed is only a few μm/s (*Talà et al., 2019*). The overnight culture of *P. aeruginosa* PA14 *flgK*::Tn5 harboring the pMHLB plasmid (with P$_{lasB}$-*gfp*(ASV), as described above), whose GFP(ASV) expression was sufficiently bright and uniform along canals for tracking purposes, was mixed with the overnight culture of PA14 *flgK*::Tn5 at a ratio of 1:100. 1 μL of this mixture was inoculated at the center of an M9DCAA agar plate, and the plate was incubated for 20 hr at 30°C and >~95% relative humidity. The fluorescent bacteria were imaged on an upright microscope (ECLIPSE Ni-E; Nikon) via a ×40 objective (N.A. 0.75) and an FITC filter cube (excitation: 482/35 nm; emission: 536/40 nm; dichroic: 506-nm-long pass filter; FITC-3540C-000, Semrock Inc), with the excitation light provided by a mercury precentered fiber illuminator (Intensilight, Nikon), and recordings were made with an sCMOS camera (Andor Neo 5.5; Andor Technology).

Due to the ultrahigh speed (up to hundreds μm/s) of cells, resolving the position of tracers (cells) frame by frame requires a very short exposure time (<~10 ms) per frame, but such a short exposure time could not yield sufficient photon count to distinguish cells from the background. To overcome this difficulty, we recorded cell trajectories during a long exposure time (0.5~5 s) (*Figure 3D*) and computed the average speed of cells during the exposure time. This way allowed us to reduce the error in speed measurement due the ultrafast movement of cells. Fifty such recordings were made at each position of the canal to yield sufficient number of cell trajectories. All recordings at a specific position were completed within 5 min from placing the agar plate on microscope stage in order to minimize the effect of evaporation. The images were imported into MATLAB (R2014b, The Math-Works; Natick, MA). For each trajectory, five points (including the two ends) were selected manually using datacursormode function in MATLAB user interface. The selected points were fitted by quadratic spline interpolation (splineinterp function in MATLAB Curve Fitting Toolbox). The length of the fitted curve was taken as the length of the trajectory, and the average speed of the cell was then calculated as the trajectory length divided by the exposure time. In each measurement, the position of field of view relative to the canal tip was determined by moving the motorized stage of the microscope, with an uncertainty of ~1 mm.

## Numerical simulation of flows in a simplified 3D canal geometry

To estimate the magnitude of surface tension gradient driving flows in canals, we performed a finite-element simulation *of* Navier–Stokes equation in a simplified 3D canal geometry using the software COMSOL Multiphysics (COMSOL, Stockholm, Sweden) (*Figure 4A*). The 3D simulation domain of the modeled canal was a liquid column with a length of 200 µm and *a* half-ellipse cross-sectional profile (the major and minor axis being 150 µm and 10 µm, respectively). We assumed a free-slip boundary condition at the upper interface of the domain (liquid–air interface) and non-slip boundary condition at the lower boundary (liquid–solid interface). A surface tension gradient was loaded at the liquid–air interface, while the pressure at both ends of the simulation domain was set as zero. The fluid viscosity was set as 0.012 Pa · s, according to a rheological measurement of *P. aeruginosa* colony extracts (*Fauvart et al., 2012*). The simulation was performed with a fine finite-element mesh. Final results of stationary solution of Navier–Stokes equation in the simulation domain were imported into MATLAB for data processing. The surface and the bulk flow speed profiles were obtained by averaging the flow speed in regions >50 µm away from both ends.

## Isolation, staining, and imaging of OMVs

To examine whether directed fluid flows in canals can facilitate long-range transport of OMVs produced by *P. aeruginosa*, we isolated OMVs from bacterial culture by centrifugation, labeled them fluorescently, and loaded the OMV dispersion into canals by microinjection. We isolated and stained *P. aeruginosa* OMVs using the following procedures. 200 mL of PA14 *flgK*::Tn5 overnight culture grown in M9DCAA medium was centrifuged at 5000 × *g* for 5 min in 50 mL centrifuge tubes. The pellet and the supernatant (containing OMVs) were collected and stored separately. The supernatant was forced to pass through a 0.45 µm syringe filter, yielding a raw OMV dispersion with all particulate matter greater than 0.45 µm in size removed. The raw OMV dispersion was further purified and concentrated by centrifugation at 3000 × *g* for 5 min in centrifugal devices with a cut-off molecular weight of 100 kDa (Microsep, Pall Corporation), yielding ~500 µL final OMV dispersion in phosphate-buffered saline (PBS; pH = 7.0). The obtained OMV dispersion was then transferred to a glass test tube for fluorescent staining with the membrane dye FM 1-43FX (Cat# F35355; Thermo Fisher Scientific). 5 µL of FM 1-43FX stock solution (1 mg/mL in DMSO) was added to the test tube and staining was allowed to proceed for 1 hr in a shaker (30°C and 180 rpm). To remove unreacted dyes, the volume of OMV dispersion in test tube was adjusted to 5.5 mL with PBS, and ~1/3 of the stored bacterial pellet was added to the test tube in order to absorb the unreacted dyes; the test tube was replaced in the shaker (30°C and 180 rpm) and incubated for 5 min, followed by centrifugation at 3000 × *g* for 5 min. The supernatant (containing fluorescently labeled OMVs) was again adjusted to a volume of ~5.5 mL with PBS and the procedures described above were repeated until using up the stored cell pellet. The resulting supernatant was forced to pass through a 0.45 µm syringe filter to remove cells, and then was purified and concentrated by centrifugation with the 100 kDa centrifugal devices as described above, yielding ~200 µL fluorescently labeled OMV dispersion in PBS. The size distribution of OMVs isolated this way was characterized by a particle sizer (NanoSight LM10, Malvern Instruments). The OMVs had a size ranging from ~30 nm to ~600 nm, with a mean size of ~150 nm (*Appendix 1—figure 5*).

Microinjection of OMVs into canals was performed using an XYZ micromanipulator (M-562-XYZ, Newport Corporation) and a 100 µL microsyringe with a glass micropipette (20 µm in diameter, fabricated by hand-pulling a 0.5 mm glass capillary tube) attached to it. The setup was installed by the side of the stage of an inverted microscope (Nikon Ti-E), where the Petri dish was placed. Prior to microinjection, fluorescently labeled OMV dispersion as described above was loaded into the microsyringe, and the lid of the Petri dish was removed. Under the microscope via a ×4 dark phase objective (N.A. 0.13), the glass micropipette was positioned to be just above the center of a canal at ~15 mm from the canal tip, and approximately 200 nL of OMV dispersion was dispensed to the canal by pushing the microsyringe. Immediately following the microinjection, the lid of Petri dished was replaced, and the transport of OMVs in the canal was imaged using a ×10 objective (N.A. 0.25) and a TRITC filter set (excitation: 535/50 nm; emission: 610/75 nm; dichroic: 565 nm long pass filter; Semrock) with the excitation light provided by a mercury precentered fiber illuminator (Intensilight, Nikon). Recordings were made with an sCMOS camera (Andor Zyla 4.2) with an exposure time of 0.1 s (for taking real-time videos as shown in *Figure 5—video 1*) or 1 s (for acquiring OMV trajectories as shown in *Figure 5C*). The final distribution of dispensed OMVs along the canal as shown in *Figure 5A and B* was imaged

via a ×4 objective (N.A. 0.13) with an exposure time of 1 s and the images at different locations were stitched by the software NIS Elements *AR*.

## Interaction between *S. aureus* and *P. aeruginosa* colonies

We used *S. aureus* cells harboring a plasmid with constitutive GFP expression (**Toledo-Arana et al., 2005**), so that the cell mass in *S. aureus* colonies can be quantified by GFP fluorescence during the interaction with *P. aeruginosa*. Overnight cultures of *S. aureus* and *P. aeruginosa* PA14 were inoculated at different sides of an M9DCAAP agar plate. After incubating at 30°C for ~20 hr, the *P. aeruginosa* colony branches were approaching *S. aureus* colonies, and then we moved the plate onto the stage of an inverted microscope (Nikon Ti-E) for imaging. Phase-contrast images were acquired with a ×4 phase-contrast objective (N.A. 0.13). Fluorescence images were acquired with a ×10 objective (N.A. 0.25) and a FITC filter set (excitation: 482/35 nm; emission: 536/40 nm; dichroic: 506 nm long pass filter; Semrock), with the excitation light provided by an LED light source (X-Cite XLED1, Excelitas Technologies Corp.). To compute fluorescence count in the images, the background (acquired at a region with no cells) was first subtracted from the fluorescent images. To correct for the inhomogeneity of excitation illumination, the illumination intensity field was acquired by taking fluorescence images of an ~100 μm thick 0.55% agar pad infused with 10 μg/ml calcein (C0875, Sigma-Aldrich) that was placed on agar surface under the same imaging setup, and then the background-subtracted fluorescence images was divided by the illumination intensity field. The agar pad was made by solidifying molten agar between two cover glasses. *S. aureus* colony biomass was measured by the total background-corrected fluorescence count in the area originally occupied by the colony prior to contact with *P. aeruginosa*.

## Acknowledgements

We thank Roberto Kolter (Harvard University) and George A O'Toole (Dartmouth College) for providing the bacterial strains; Peter Greenberg (University of Washington) for helpful comments and for providing bacterial strains; To Ngai and Hang Jiang (CUHK) for assistance with surface tension measurement; Shannon Au (CUHK) for help with membrane vesicle isolation; and Bo Zheng and Qi Liu (CUHK) for help with PDMS microfluidic chip fabrication. We also thank Qihui Hou for helpful discussion and information on rhamnolipid control of colony patterns. This work was supported by the Ministry of Science and Technology Most China (no. 2020YFA0910700 to YW), the Research Grants Council of Hong Kong SAR (RGC ref no. 14306820, 14307821, RFS2021-4S04 and CUHK Direct Grants; to YW), Guangdong Natural Science Foundation for Distinguished Young Scholar (no. 2020B1515020003, to LY), and Guangdong Basic and Applied Basic Research Foundation (no. 2019A1515110640, to YZ).

## Additional information

### Funding

| Funder | Grant reference number | Author |
|---|---|---|
| Ministry of Science and Technology of the People's Republic of China | No. 2020YFA0910700 | Yilin Wu |
| Research Grants Council, University Grants Committee | 14306820 | Yilin Wu |
| Research Grants Council, University Grants Committee | 14307821 | Yilin Wu |
| Research Grants Council, University Grants Committee | RFS2021-4S04 | Yilin Wu |

| Funder | Grant reference number | Author |
| --- | --- | --- |
| Research Grants Council, University Grants Committee | CUHK Direct Grants | Yilin Wu |
| Guangdong Natural Science Foundation | No. 2020B1515020003 | Liang Yang |
| Guangdong Basic and Applied Basic Research Foundation | No. 2019A1515110640 | Yingdan Zhang |

The funders had no role in study design, data collection and interpretation, or the decision to submit the work for publication.

### Author contributions

Ye Li, Shiqi Liu, Data curation, Formal analysis, Investigation, Methodology; Yingdan Zhang, Funding acquisition, Methodology; Zi Jing Seng, Methodology; Haoran Xu, Data curation, Investigation; Liang Yang, Resources, Supervision, Funding acquisition, Methodology; Yilin Wu, Conceptualization, Resources, Formal analysis, Supervision, Funding acquisition, Investigation, Methodology, Writing - original draft, Project administration, Writing - review and editing

### Author ORCIDs

Haoran Xu (iD) http://orcid.org/0000-0001-9613-297X
Yilin Wu (iD) http://orcid.org/0000-0002-0392-2137

### Decision letter and Author response

Decision letter https://doi.org/10.7554/eLife.79780.sa1
Author response https://doi.org/10.7554/eLife.79780.sa2

## Additional files

### Supplementary files

• MDAR checklist

### Data availability

All data are available in the main text or the appendix.

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

## Appendix 1

## Mathematical model for the dynamics of surfactant distribution

A key component of our model for the spatial-temporal dynamics of surfactant distribution is biosurfactant exchange between the liquid–air interface and the bulk phase. The kinetics of biosurfactant exchange between the two phases could be learned from a relation between the bulk concentration and the surface concentration of biosurfactant at steady state.

Based on our measurement of surface tension at different rhamnolipid concentrations (**Appendix 1—figure 3**), the relation between the steady-state surface tension $\gamma_{ss}$ and bulk concentration of biosurfactant $c$ can be well described by an exponential decay:

$$\gamma_{ss}\left(c\right) = \Pi_{max} exp\left(-c/c_s\right) + \gamma_\infty \tag{S1}$$

In **Equation S1** $\Pi_{max} = 41.6\ \mathrm{mN \cdot m^{-1}}$ is the maximal decrease of surface tension; $c_s = 12.9\ \mathrm{mg \cdot L^{-1}}$ is the characteristic concentration of rhamnolipids; $\gamma_\infty = 30.6\ \mathrm{mN \cdot m^{-1}}$ is the saturated surface tension of rhamnolipid solutions.

On the other hand, the surface tension directly depends on the surface concentration of surfactants ($\Gamma$). Such a relation is well fitted by a quadratic exponential decay (**Hanyak et al., 2012**):

$$\gamma\left(\Gamma\right) = \Pi_{max} exp\left(-A\Gamma^2/\Gamma_c^2\right) + \gamma_\infty \tag{S2}$$

In **Equation S2**, $\Gamma_c$ is the characteristic surface concentration of surfactant, A, is a parameter relating bulk surfactant concentration to the steady-state surface density of the surfactant.

Comparing **Equation S2** to **Equation S1**, we find that $c \sim A\Gamma^2$ at equilibrium. This means that the surface concentration of rhamnilipids does not saturate but varies with the bulk concentration. To account for this behavior of rhamnilipids, we model the rate of biosurfactant transfer ( $r_{net}$) from the bulk phase to the interface as follows:

$$r_{net} \propto \frac{c}{c_s} - A\frac{\Gamma^2}{\Gamma_c^2} = k\left(\frac{c}{c_s} - A\frac{\Gamma^2}{\Gamma_c^2}\right) \tag{S3}$$

In **Equation S3**, $k$ is a reaction rate constant. To simplify the analysis, we introduced the dimensionless bulk concentration $c_n = c/c_s$ and the dimensionless surface concentration $\Gamma_n = \Gamma/\Gamma_c$ of rhamnilipids, respectively, and rewrite **Equation S3**:

$$r_{net} = k\left(c_n - A\Gamma_n^2\right) \tag{S4}$$

The spatial-temporal dynamics of the bulk and the interface biosurfactant concentration $c_n$ and $\Gamma_n$ are described by the following coupled differential equations:

$$\frac{\partial c_n}{\partial t} = \nabla \cdot \left(D_c \nabla c_n\right) - k\left(c_n - A\Gamma_n^2\right) + \alpha_R \rho \frac{N}{K_N + N}\frac{B^m}{K_B^m + B^m} \tag{S5}$$

$$\frac{\partial \Gamma_n}{\partial t} = \nabla \cdot \left(D_\Gamma \nabla \Gamma_n\right) - \nabla \cdot \left[\eta_\Gamma \Gamma_n \nabla \gamma\left(\Gamma_n\right)\right] + k\left(c_n - A\Gamma_n^2\right) \tag{S6}$$

Here, $D_\Gamma$, $D_c$, $\eta_\Gamma$, $k$, $A$, $K_N$, $\alpha_R$, $K_B$, $K_N$, and $m$ are constant parameters whose definitions and values are described in **Appendix 1—table 2**. **Equation S5** describes the variation of $c_n$ due to three processes, namely, biosurfactant diffusion in the bulk phase, biosurfactant exchange between the liquid–air interface and the bulk phase (see **Equation S4**), and biosurfactant production. The biosurfactant production term is proportional to bacterial density ($\rho$) and bacterial metabolic activity (which is limited by nutrient concentration $N$ and modeled as a Hill function [**Cao et al., 2016**], $N/\left(K_N + N\right)$ ), and regulated by the QS activity (modeled as a Hill function of auto-inducer concentration $B$ with Hill coefficient $m$; **Cao et al., 2016**); $\rho$, $N$, and $B$ follow another set of differential equations to be described below. **Equation S6** describes the variation of $\Gamma_n$ due to biosurfactant diffusion at the interface, advective biosurfactant transport by Marangoni flows, as well as biosurfactant exchange between the liquid–air interface and the bulk phase. Biosurfactants adsorbed to the liquid–air interface will be carried by liquid flows in canals, hence giving rise to the advective transport term in **Equation S6**. The speed of Marangoni flows in the advective transport

term is proportional to surface tension gradient $\nabla\gamma\left(\Gamma_n\right)$ (*de Gennes et al., 2003*). Biosurfactant concentration in the bulk phase is assumed to be unaffected by Marangoni flows because the volume of the bulk phase is much greater than that of the canals in our experiments.

The spatial-temporal dynamics of $N$, $\rho$, and $B$ are described by the following coupled differential equations:

$$\frac{\partial N}{\partial t} = \nabla \cdot \left(D_N \nabla N\right) - \beta \frac{N}{K_N + N}\rho \tag{S7}$$

$$\frac{\partial \rho}{\partial t} = \nabla \cdot \left(D_\rho \nabla \rho\right) - \nabla \cdot \left[\eta_\rho \rho \nabla\gamma\left(\Gamma_n\right)\right] + \beta \frac{N}{K_N + N}\rho \frac{1}{1 + \lambda \frac{B^m}{K_B^m + B^m}} \tag{S8}$$

$$\frac{\partial B}{\partial t} = \nabla \cdot \left(D_B \nabla B\right) + \alpha_B \rho \frac{B^m}{K_B^m + B^m} + \chi\rho - d_B B \tag{S9}$$

Here, $D_N$, $D_\rho$, $D_B$, $\beta$ , $\eta_\rho$, $\lambda$ , $\alpha_B$, $\chi$, and $d_B$ are constant parameters whose definitions and values are described in *Appendix 1—table 2*. The variation of the nutrient concentration field $N$ (*Equation S7*) is due to diffusion and consumption, with the nutrient consumption term being proportional to bacterial density $\rho$ and bacterial metabolic activity $N/\left(K_N + N\right)$ as described above for *Equation S5*. The variation of the bacterial density field $\rho$ (*Equation S8*) is due to three processes, namely, diffusion, advective transport by Marangoni flows, and bacterial growth; the bacterial growth term is proportional to bacterial density $\rho$ and bacterial metabolic activity, and is repressed by the QS-regulated biosurfactant production due to the associated metabolic cost. The variation of the QS auto-inducer concentration field $B$ is described in *Equation S9*. It includes four components, a diffusion term, a production term proportional to bacterial density $\rho$ and regulated by QS activity via a Hill function with Hill coefficient $m$, a basal production term proportional to $\rho$ (to ensure that autoinduction can be activated), and a degradation term.

*Equations S2; S5–S9* constitute the entire model for the spatial-temporal dynamics of biosurfactant distribution. All variables in the model are dimensionless. To simplify the computation, we numerically solved *Equations S5–S9* in polar coordinate system with rotational symmetry. The computation was programmed in Fortran and complied by gFortran 9 on Linux platform. With appropriately chosen model parameters (*Appendix 1—table 2*), the magnitude of surface tension gradient was tuned to ~1000–3000 $mN \cdot m^{-2}$ , so as to match the estimated magnitude of surface tension gradient (*Figure 4A*) required to generate the observed flow speed in canals (*Figure 3E*).

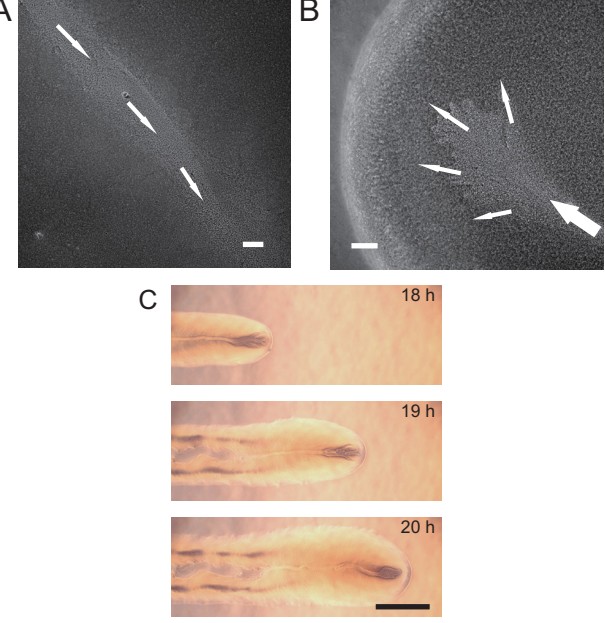

**Appendix 1—figure 1.** Open channels supporting long-range directed material transport in *P. aeruginosa* colonies. (**A**) Phase-contrast microscopy image of an open channel in a *P. aeruginosa* (PA14 *flgK*::Tn5) colony.
*Appendix 1—figure 1 continued on next page*

*Appendix 1—figure 1 continued*

Arrows indicate the flow direction. Scale bar, 100 μm. Also see ***Appendix 1—figure 1—video 1***. (**B**) Phase-contrast microscopy image taken near the tip of an open channel in a *P. aeruginosa* (PA14 *flgK*::Tn5) colony. Arrows indicate the flow direction. Scale bar, 100 μm. Also see ***Appendix 1—figure 1—video 2***. (**C**) Image sequence taken by a DSLR camera via a ×4 phase-contrast objective lens showing the development of an open channel (or canal) in a branching colony of *P. aeruginosa* (PA14 *flgK*::Tn5). Scale bar, 1 mm. Also see ***Appendix 1—figure 1—video 3***. The fluid flow in bacterial canals was sensitive to water content in the air environment and it was easily disrupted by decrease of humidity. Cells translocating along the open channels eventually settled in near the colony edge and they may contribute to colony expansion. Fluid flow in open channels on average went toward the colony edge and stopped abruptly at the very end (i.e., the tip) of an open channel, disappearing into the dense layer of cells near the edge (panel **B**; *Appendix 1—figure 1—video 2*).

**Appendix 1—figure 1—video 1**. Cellular flow in an open channel of a piliated *P. aeruginosa* (PA14 *flgK*::Tn5) colony. This real-time video is associated with ***Appendix 1—figure 1A***.
https://elifesciences.org/articles/79780/figures#app1fig1video1

**Appendix 1—figure 1—video 2.** Cellular flow near the tip of an open channel in a piliated *P. aeruginosa* (PA14 *flgK*::Tn5) colony. The flow slowed down and stopped abruptly at the tip, disappearing into the dense layer of cells near the edge of the colony. Note that the colony edge had ceased expansion and in general, canal formation does not necessarily coincide with colony expansion. This real-time video is associated with ***Appendix 1—figure 1B***.
https://elifesciences.org/articles/79780/figures#app1fig1video2

**Appendix 1—figure 1—video 3.** Zoomed-in view of a developing bacterial canal in a branching colony of piliated *P. aeruginosa* (PA14 *flgK*::Tn5). Time label shows the elapsed time from inoculation (hh:mm). The video is associated with ***Appendix 1—figure 1C***.
https://elifesciences.org/articles/79780/figures#app1fig1video3

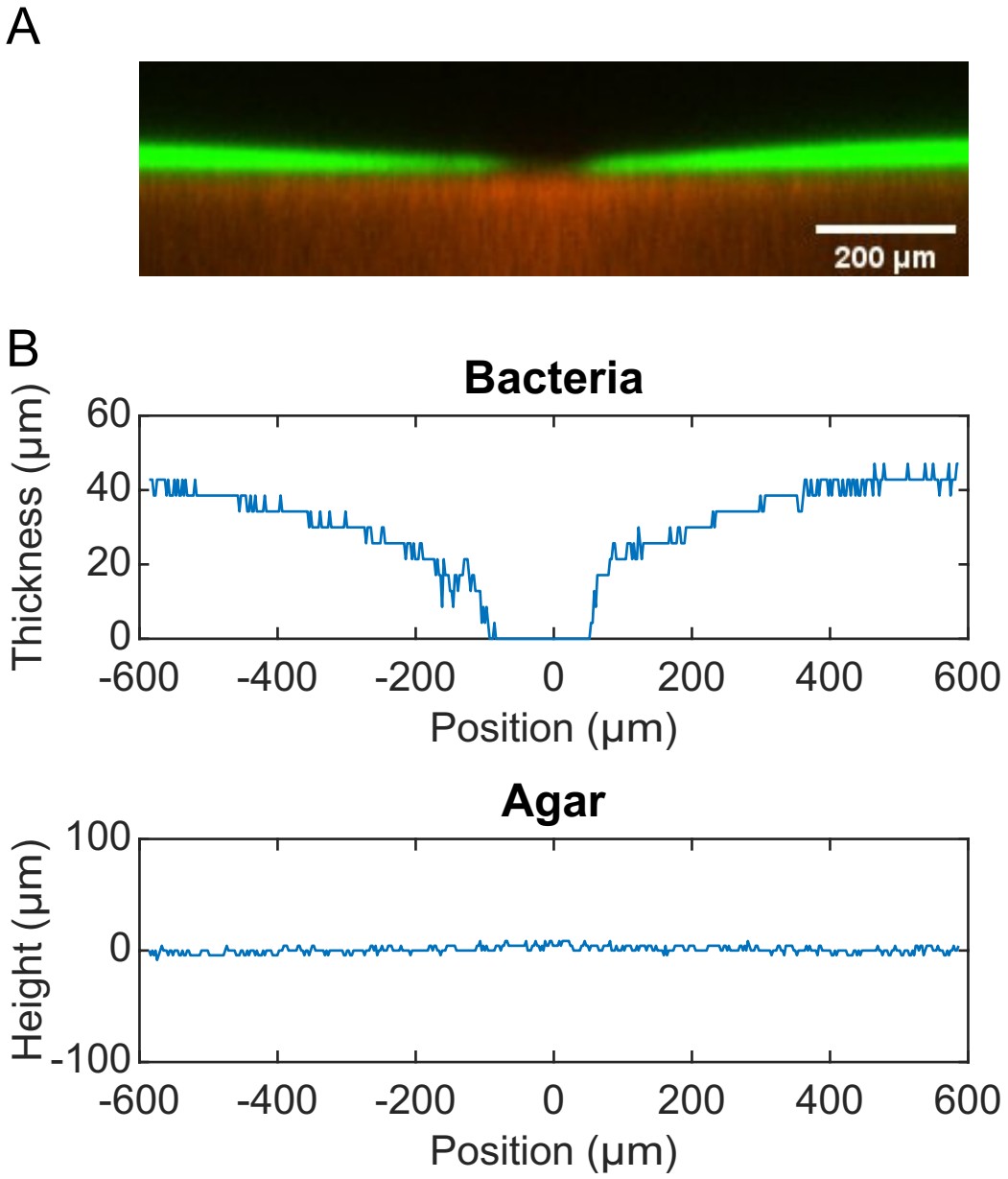

**Appendix 1—figure 2.** Height profiles of colony and agar. (**A**) Cross-sectional view of the colony (green fluorescence) and the agar underneath the colony (red fluorescence) measured by laser scanning confocal microscopy (see 'Methods'). A canal is located at the center of the image. (**B**) Thickness profile of the colony (upper panel) and height profile of the agar underneath the colony (lower panel) associated with panel (**A**). The height of agar at position 600 μm was chosen as the reference level (i.e., height = 0 μm). The thickness of the colony is low inside the canal region due to flushing of fluid flows. The height of agar is uniform, showing that canal formation is not associated with agar degradation.

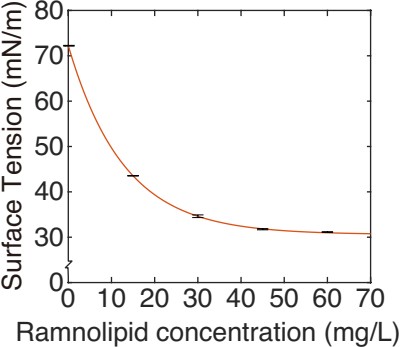

**Appendix 1—figure 3.** Surface tension of rhamnolipid solutions as a function of rhamnolipid concentration. The relation between bulk rhamnolipid concentration and the steady-state surface tension. Steady-state surface tension was measured by pendant drop assay with a commercial contact-angle meter (OCA25, DataPhysics, Germany). The rhamnolipid solutions were prepared by dissolving rhamnolipids (in solid form) in M9DCAA medium. Error bars indicate the standard deviation (N=5). The line is a fit of data points to an exponential decay function (*Equation 1* of Appendix 1 – text).

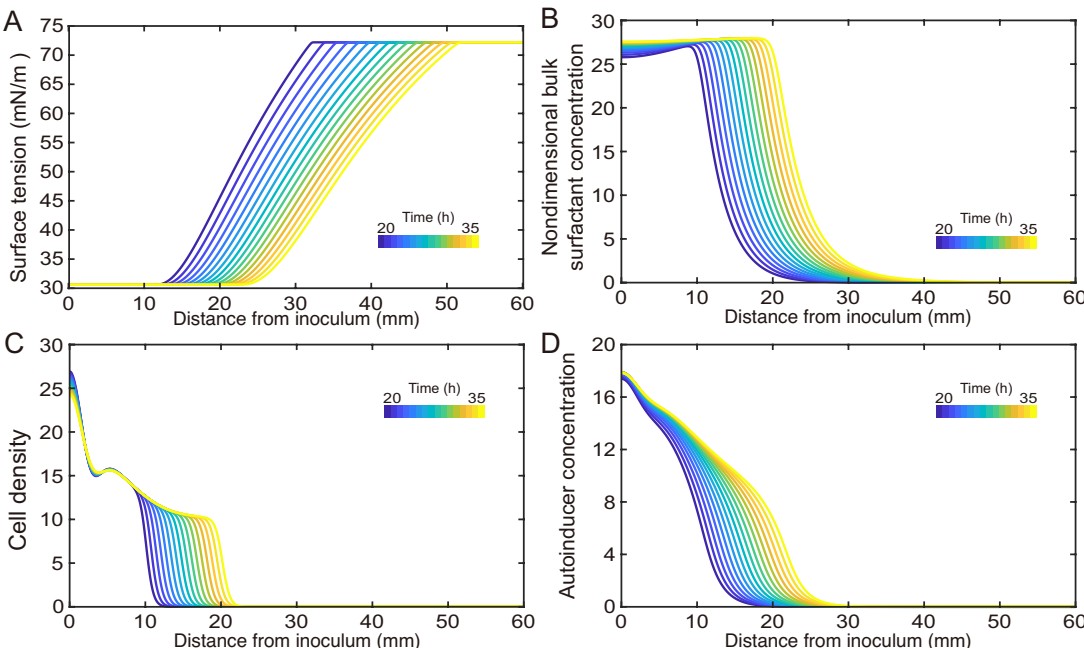

**Appendix 1—figure 4.** Spatial-temporal dynamics of key model variables related to the development of surface tension gradient in canals. This figure is associated with *Figure 4*. The spatial-temporal dynamics of the following variables used in the model presented in main text and in the text of Appendix 1 were plotted: surface tension (**A**); bulk biosurfactant concentration (**B**); bacterial density (**C**); autoinducer concentration (**D**).

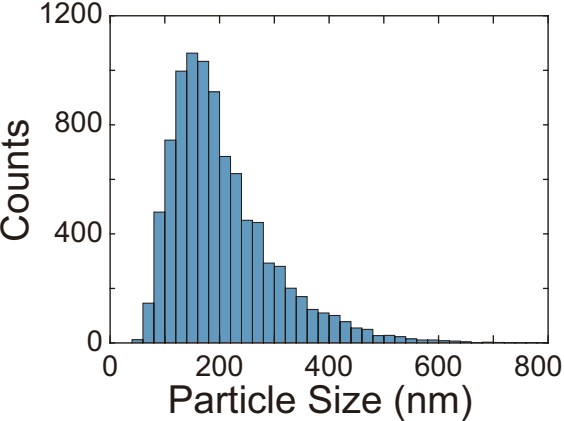

**Appendix 1—figure 5.** Size distribution of outer membrane vesicles (OMVs). The OMVs were derived from canal-forming *P. aeruginosa* (PA14 *flgK::*Tn5) and the size distribution was measured by a particle sizer (NanoSight LM10, Malvern Instruments). See 'Methods.'

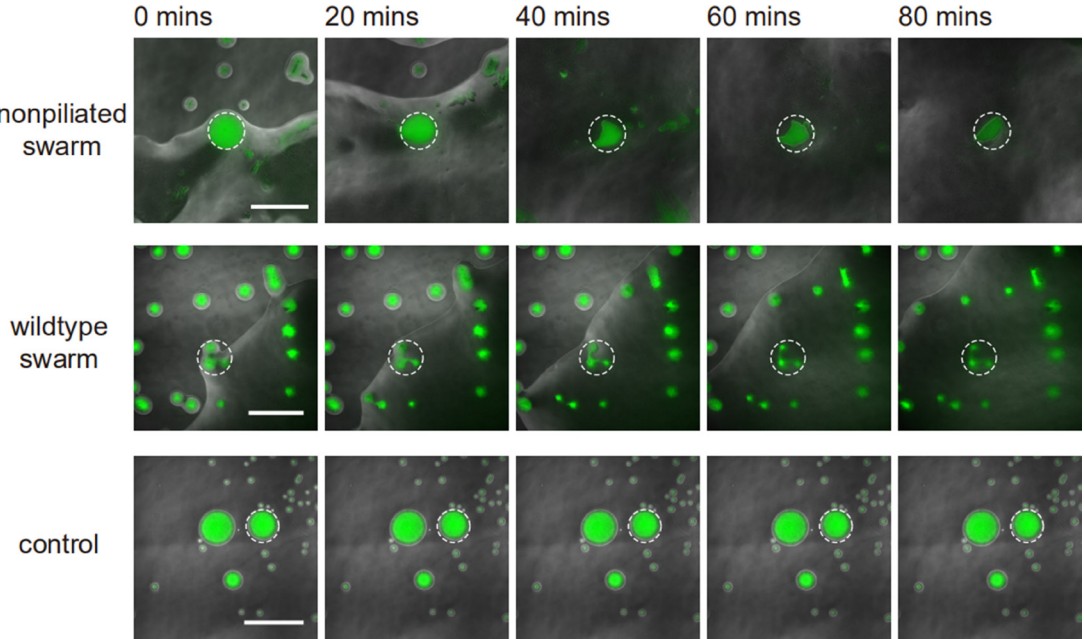

**Appendix 1—figure 6.** Interaction between *S. aureus* and *P. aeruginosa* PA14 colonies. This figure is associated with **Figure 5D and E**. To compare with the effect of potential OMV transport due to flagellar motility, we co-cultured swarms of flagellated *P. aeruginosa* PA14 with *S. aureus* colonies under the same conditions as used in **Figure 5D**. The image sequences show the temporal variation of *S. aureus* colony biomass (green fluorescence) under the following conditions: top, in contact with the swarming colony of PA14 Δ*pilB* mutant; middle, in contact with the swarming colony of wildtype PA14; bottom, control experiment in the absence of *P. aeruginosa*. Both PA14 Δ*pilB* mutant and wildtype PA14 have flagellar motility, but neither of them can form canals. In each panel, the fluorescence image of *S. aureus* colonies (green) is superimposed onto the phase-contrast image. The dashed circles mark the region with *S. aureus* colonies for fluorescence analysis in **Figure 5E**. As shown in **Figure 5E**, *the S. aureus* colonies encountering expanding *P. aeruginosa* swarms were destructed to a lesser extent compared to the case that were irrigated by canal flows, retaining ~20% and ~50% of cell mass after 60 min of contact with swarming colonies of *P. aeruginosa* PA14 Δ*pilB* mutant and wildtype PA14, respectively.

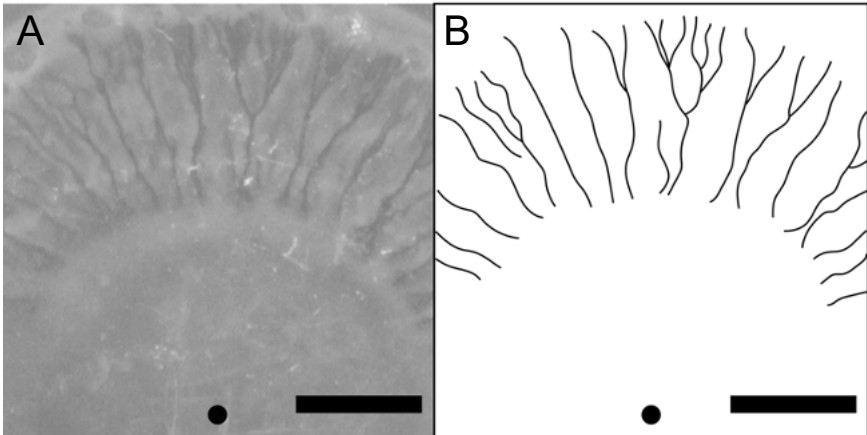

**Appendix 1—figure 7.** Canal formation in *Serratia marcescens* colonies. Overnight culture of *S. marcescens* grown in LB medium was inoculated on 0.6% Eiken agar plates and photographed in a custom-built imaging incubator made of PMMA. The temperature of the incubator was maintained at 30°C and the images of canals were photographed by a digital single-lens reflex camera. (**A**) Canal appeared ~4 hr after colony inoculation. (**B**) Sketch of the canals in panel (**A**) for better visualization. The black dot represents the inoculum. Scale bars, 500 µm.

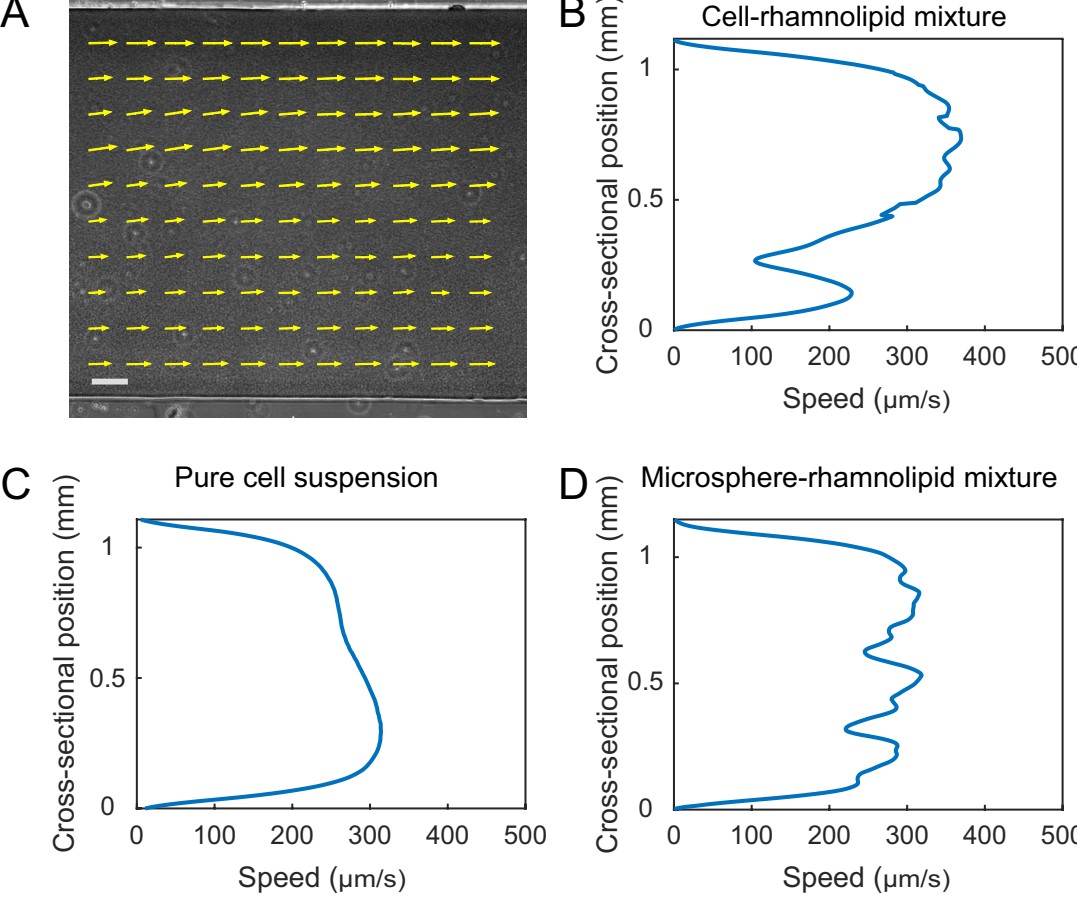

**Appendix 1—figure 8.** Shear banding of cell-rhamnolipid mixture in PDMS microfluidic channels. (**A**) Flow velocity field of cell-rhamnolipid mixture in a microfluidic channel. The microfluidic channel had a rectangular cross section (200 µm of height, 1.1 mm of width). Cells (PA14 *flgK*::Tn5; ZK3367) and rhamnolipids were mixed at a final density

*Appendix 1—table 1 Continued on next page*

*Appendix 1—figure 8 continued*

(concentration) of 1.9 × 10^10 cells/mL and 10 mg/mL, respectively, and driven by a syringe pump at a flow rate of 5 µL/min. The velocity field was measured by particle image velocimetry (PIV) analysis on phase-contrast image sequence and averaged over a time window of 5 s. The velocity field is superimposed onto one of the analyzed phase-contrast images. Arrows indicate velocity vectors. Scale bar, 100 µm. (**B**) Cross-sectional flow speed profile computed based on velocity field data shown in panel (**A**). Each data point represents local velocity averaged in a domain of 41.6 µm × 41.6 µm centered around the cross-sectional position. (**C, D**) Cross-sectional flow speed profile in microfluidic channels filled with other fluids. Data in panel (**C**) was obtained with pure cell suspension (1.9 × 10^10 cells/mL), and data in panel (**D**) with a mixture of rhamnolipids (10 mg/mL) and 1.1 µm microsphere suspension (~1.9 × 10^10 particles/mL). These fluids were driven through microfluidic channels by a syringe pump at a flow rate of 3 µL/min. The flow speed profiles were computed based on the respective time-averaged velocity field, which was obtained by PIV analysis on phase-contrast image sequence and averaged over a time window of 5 s. Each data point in (**C, D**) represents local velocity averaged in a domain of 41.6 µm × 41.6 µm centered around the cross-sectional position. See 'Methods' for details of experiments and image analysis associated with all panels.

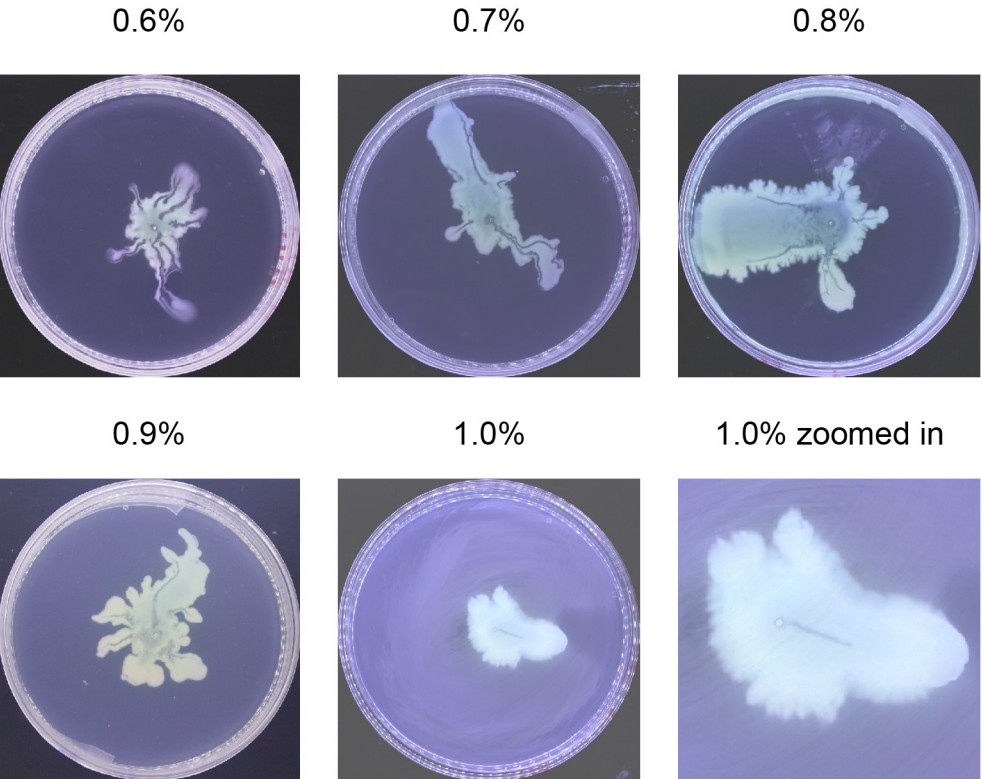

**Appendix 1—figure 9.** Representative images of canal development at different agar concentrations. The colonies were from the same batch and incubated under the same condition (30°C) for the following durations: 0.6–0.8%, 19 hr; 0.9%, 24 hr; 1.0%, 36 hr. Canals can be found in different agar (infused with M9DCAA medium) concentrations from 0.6% to 1.0%, in addition to the 0.5% agar used in the main text. Note that a canal in the colony on the 1.0% plate is located at the center (see the zoomed-in view).

**Appendix 1—table 1.** Primers.

| No. | Oligo name | Sequence 5' to 3' |
|---|---|---|
| 1 | 1-rhlA_UpF | agctcggtacccgggGGGTGATTTCCTACGGGGTG |
| 2 | 2-rhlA_UpR | CTTCGCAGGTCAAGGGTTCACCGCATTTCACACCTCCCAA |
| 3 | 3-rhlA_DownF | TTGGGAGGTGTGAAATGCGGTGAACCCTTGACCTGCGAAG |
| 4 | 4-rhlA_DownR | cgacggccagtgccaCCGTACTTCTCGTGAGCGAT |
| 5 | rhlA_F | GACAAGTGGATTCGCCGCA |
| 6 | rhlA_R | TTGAACTTGGGGTGTACCGG |
| 7 | rhlAGENE_F | GGTCAATCACCTGGTCTCCG |
| 8 | rhlAGENE_R | GCTGATGGTTGCTGGCTTTC |
| 9 | Pk18-F | TGCTTCCGGCTCGTATGTTG |
| 10 | Pk18-R | GCGAAAGGGGGATGTGCTG |

**Appendix 1—table 2.** Simulation parameters.

| Parameter | Description | Value | Unit | Robust range * |
|---|---|---|---|---|
| $\alpha_R$ | Biosurfactant synthesis rate | $2 \times 10^{-3}$ | s$^{-1}$ | $5 \times 10^{-4}$-$1 \times 10^{-2}$ |
| $\beta$ | Bacterial growth rate | $6 \times 10^{-4}$ | s$^{-1}$ | $3 \times 10^{-4}$-$1 \times 10^{-3}$ |
| $K_N$ | Nutrient concentration for half maximal metabolic activity | 1 | Nondimensio-nal | N/A |
| $K_B$ | Half-activation threshold of QS auto-inducer | 1 | Nondimensio-nal | N/A |
| $\alpha_B$ | Auto-inducer synthesis rate | $2 \times 10^{-4}$ | s$^{-1}$ | $1 \times 10^{-4}$-$1 \times 10^{-3}$ |
| $d_B$ | Auto-inducer degradation rate | $2 \times 10^{-4}$ | s$^{-1}$ | $0$-$2 \times 10^{-4}$ |
| $\chi$ | Auto-inducer basal production rate | $2 \times 10^{-6}$ | s$^{-1}$ | $1 \times 10^{-6}$-$2 \times 10^{-5}$ |
| $m$ † | Hill coefficient of QS regulation | 2 | | N/A |
| $\lambda$ | Parameter of growth-repression due to QS-regulated biosurfactant production | 0.3 | Nondimensio-nal | 0–0.5 |
| $k$ ‡ | Reaction rate constant for biosurfactant transfer between the bulk phase and the interface | $2 \times 10^{-5}$ | s$^{-1}$ | N/A |
| $A$ § | Ratio between bulk surfactant concentration and square of the steady-state surface density | 6.125 | Nondimensio-nal | 2–10 |
| $\eta_\rho$‡ | Parameter of advective bacterial transport | $5 \times 10^{-11}$ | (mN · m$^{-2}$)$^{-1}$s$^{-1}$ | N/A |
| $\eta_r$‡ | Parameter of advective biosurfactant transport | $2.5 \times 10^{-10}$ | (mN · m$^{-2}$)$^{-1}$s$^{-1}$ | N/A |
| $D_c$ | Diffusion coefficient of biosurfactant in bulk phase | $1 \times 10^{-9}$ | m$^2$s$^{-1}$ | $0$-$1 \times 10^{-9}$ |
| $D_\Gamma$ | Diffusion coefficient of biosurfactant at liquid–air interface | $5 \times 10^{-11}$ | m$^2$s$^{-1}$ | $0$-$1 \times 10^{-10}$ |
| $D_\rho$ | Diffusion coefficient of bacteria | $5 \times 10^{-11}$ | m$^2$s$^{-1}$ | $0$-$5 \times 10^{-11}$ |
| $D_B$ | Diffusion coefficient of auto-inducers | $1 \times 10^{-9}$ | m$^2$s$^{-1}$ | $5 \times 10^{-10}$-$1 \times 10^{-9}$ |
| $D_N$ | Diffusion coefficient of nutrient | $1 \times 10^{-9}$ | m$^2$s$^{-1}$ | $0$-$2 \times 10^{-9}$ |

*Appendix 1—table 2 Continued on next page*

Appendix 1—table 2 Continued

| Parameter | Description | Value | Unit | Robust range * |
|---|---|---|---|---|

*The results shown in **Figure 4D and E** are robust to variation of parameters within the indicated range.

†The value of this parameter was taken from **Payne et al., 2013**.

‡These parameters are chosen to yield the width of the plateau of surface tension gradient distribution (**Figure 4D**) being ~25 mm (matching the experimental results).

§The value of this parameter was taken from **Hanyak et al., 2012**.

