## [Editor Report]

Congratulations on a very nice study. We see the discovery of the formation of large self-organized canals within bacterial colonies as a fascinating phenomenon. Furthermore, your thorough analysis describing the capacity of bacteria to employ these canals for rapid long-distance intercellular transportation has major implications for our understanding of bacterial cooperation. We are delighted to have the opportunity to present your fundamental findings to the scientific community.

---

## [Decision Letter]

**Decision letter after peer review:**

Thank you for submitting your article "Self-organized canals enable long range directed material transport in bacterial communities" for consideration by *eLife*. We apologize for the delayed response. Your article has been reviewed by 2 peer reviewers, and the evaluation has been overseen by a Reviewing Editor and Naama Barkai as the Senior Editor. The reviewers have opted to remain anonymous.

Essential revisions:

1. Please highlight the shortage of using flagellum mutants and compare your findings with that of *P. aeruginosa* cells imaged in situ, as indicated by Reviewer #1.

2. Please consider focusing more on the physics of the system and its potential for synthetic systems in the Introduction and Discussion, rather than on multicellularity.

---

## [Author Response]

Essential revisions:1. Please highlight the shortage of using flagellum mutants and compare your findings with that of *P. aeruginosa* cells imaged in situ, as indicated by Reviewer #1.

We have revised the third paragraph of Discussion section to limit the generality of our findings in clinical or ecological settings (lines 441-458). Results of imaging *P. aeruginosa* cells in situ in sputum samples from cystic fibrosis patients are compared, and the shortage of using flagellum mutants is highlighted.

2. Please consider focusing more on the physics of the system and its potential for synthetic systems in the Introduction and Discussion, rather than on multicellularity.

We have thoroughly revised the Introduction and Discussion according to this comment (revising/adding texts in lines 58-64, 93-99, 429-439; deleting texts related to multicellularity in Introduction/Discussion).